# Bottom-up and top-down computations in word- and face-selective cortex

Kendrick N Kay[1]*, Jason D Yeatman[2,3]*

[1]Center for Magnetic Resonance Research, Department of Radiology, University of Minnesota, Minneapolis, United States; [2]Institute for Learning and Brain Sciences, University of Washington, Seattle, United States; [3]Department of Speech and Hearing Sciences, University of Washington, Seattle, United States

**Abstract** The ability to read a page of text or recognize a person's face depends on category-selective visual regions in ventral temporal cortex (VTC). To understand how these regions mediate word and face recognition, it is necessary to characterize how stimuli are represented and how this representation is used in the execution of a cognitive task. Here, we show that the response of a category-selective region in VTC can be computed as the degree to which the low-level properties of the stimulus match a category template. Moreover, we show that during execution of a task, the bottom-up representation is scaled by the intraparietal sulcus (IPS), and that the level of IPS engagement reflects the cognitive demands of the task. These results provide an account of neural processing in VTC in the form of a model that addresses both bottom-up and top-down effects and quantitatively predicts VTC responses.

## Introduction

How does visual cortex work? One approach to answering this question consists in building functional models that characterize the computations that are implemented by neurons and their circuitry (*Hubel and Wiesel, 1963*; *Heeger et al., 1996*). This approach has been fruitful for the front end of the visual system, where relatively simple image computations have been shown to characterize the spiking activity of neurons in the retina, thalamus, and V1 (*Carandini et al., 2005*; *Wu et al., 2006*). Based on this pioneering work in electrophysiology, researchers have extended the modeling approach to characterize responses in human visual cortex, as measured by functional magnetic resonance imaging (fMRI) (*Wandell, 1999*; *Dumoulin and Wandell, 2008*; *Kay et al., 2008*).

Models of early visual processing have been able to offer accurate explanations of low-level perceptual functions such as contrast detection (*Ress et al., 2000*; *Ress and Heeger, 2003*) and orientation discrimination (*Bejjanki et al., 2011*). However, these models are insufficient to explain high-level perceptual functions such as the ability to read a page of text or recognize a face. These abilities are believed to depend on category-selective regions in ventral temporal cortex (VTC), but the computations that give rise to category-selective responses are poorly understood.

The goal of the present study is to develop a model that predicts fMRI responses in high-level visual cortex of human observers while they perform different cognitive tasks on a wide range of images. We seek a model that is fully computable—that is, a model that can operate on any arbitrary visual image and quantitatively predict BOLD responses and behavior (*Kay et al., 2008*; *Huth et al., 2012*; *Khaligh-Razavi and Kriegeskorte, 2014*; *Yamins et al., 2014*). Achieving this goal requires four innovations: First, we need to develop a forward model that characterizes the relationship between visual inputs and the BOLD response in word- and face-selective cortex. Second, we need to dissociate bottom-up stimulus-driven effects from modulation by top-down cognitive processes and characterize how these processes alter the stimulus representation. Third, we need to localize

*For correspondence: kay@umn.edu (KNK); jyeatman@uw.edu (JDY)

Competing interests: The authors declare that no competing interests exist.

the source of the top-down effects and integrate bottom-up and top-down computations into a single consolidated model. Finally, the neural computations should be linked to the measured behavior of the visual observer. In this study, we make progress on these four innovations and develop a model that characterizes bottom-up and top-down computations in word- and face-selective cortex.

## Results

### VTC responses depend on both stimulus properties and cognitive task

Ventral temporal cortex (VTC) is divided into a mosaic of high-level visual regions that respond selectively to specific image categories, and are believed to play an essential role in object perception (*Kanwisher, 2010*; *Dehaene and Cohen, 2011*). We focus on two specific VTC regions, the visual word form area (VWFA), which selectively responds to words (*Cohen et al., 2000*, *2002*; *Wandell et al., 2012*), and the fusiform face area (FFA), which selectively responds to faces (*Kanwisher et al., 1997*; *Grill-Spector and Weiner, 2014*).

We measured blood oxygenation level dependent (BOLD) responses to a set of carefully controlled images while manipulating the cognitive task that the subjects performed on the stimuli. The first task was designed to minimize the influence of cognitive processes on sensory processing of the stimulus. Subjects performed a demanding perceptual task on a small dot (0.12° × 0.12°) presented at fixation. In this *fixation task*, the presented stimuli are irrelevant to the subject, and we interpret evoked activity as reflecting primarily the intrinsic, bottom-up response from VTC. We acknowledge that the fixation task may not perfectly isolate bottom-up responses. For example, high-contrast stimuli may automatically attract attention. Moreover, there are other potential interpretations of the fixation task: for example, allocating attention to the small fixation dot might engage active suppression of responses to the presented stimuli.

To a first approximation, much of the variance in the bottom-up fixation responses from VWFA and FFA is explained by the category of stimulus (*Figure 1d*, red lines). However, we find that responses are not invariant to low-level properties of the stimulus: both image contrast and phase coherence modulate response amplitudes. For example, the response to a word in VWFA is 2.4 times stronger when the word is presented at 100% contrast as compared to 3% contrast. These

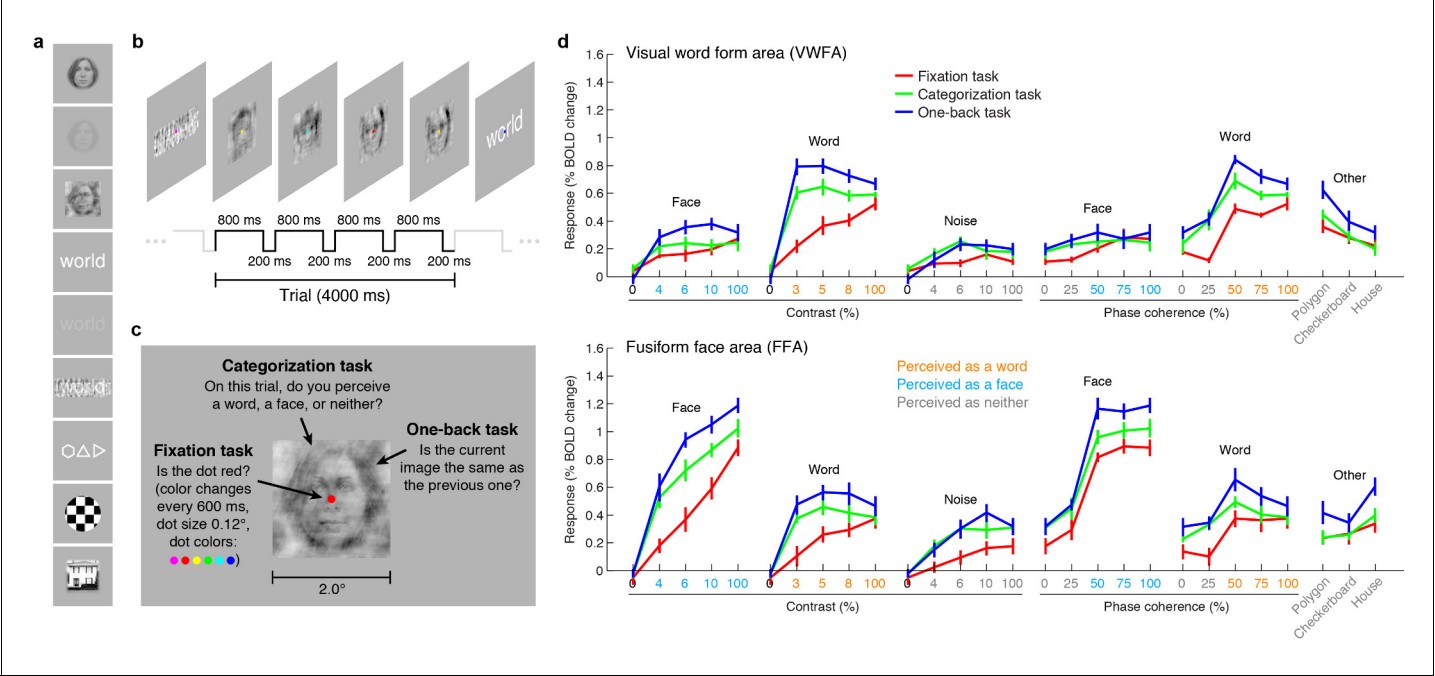

**Figure 1.** VTC responses depend on both stimulus properties and cognitive task. (a) *Stimuli.* Stimuli included faces, words, and noise patterns presented at different contrasts and phase-coherence levels, as well as full-contrast polygons, checkerboards, and houses. (b) *Trial design.* Each trial consisted of four images drawn from the same stimulus type. (c) *Tasks.* On a given trial, subjects performed one of three tasks. (d) *Evoked responses in VWFA (top) and FFA (bottom) for different stimuli and tasks.* Color of *x*-axis label indicates the perceived stimulus category as reported by the subjects. Error bars indicate bootstrapped 68% CIs.

The following figure supplement is available for figure 1:

**Figure supplement 1.** Comprehensive summary of fMRI measurements.

bottom-up effects (see also *Rainer et al., 2001*; *Avidan et al., 2002*; *Yue et al., 2011*; *Nasr et al., 2014*) may be somewhat surprising given that theories of word recognition generally posit that the VWFA response is invariant to low-level features (*Dehaene and Cohen, 2007*, *2011*; *Price and Devlin, 2011*). In fact, it is currently debated whether the VWFA should be considered a visual area or a 'meta modal' language region (*Reich et al., 2011*; *Striem-Amit et al., 2012*). Our measurements indicate that when top-down signals are minimized, word- and face-selective cortex is sensitive to low-level image properties, and that an accurate model of the computations performed by these regions must consider not only the stimulus category but also low-level features of the stimulus.

We also measured VTC responses while subjects performed a *categorization task*, in which the subject reports the perceived category of the stimulus, and a *one-back task*, in which the subject detects consecutive repetitions of stimulus frames. Despite the presentation of identical stimuli across the three tasks, there are substantial changes in evoked VTC responses. Responses are larger for the categorization (*Figure 1d*, green lines) and one-back tasks (*Figure 1d*, blue lines) compared to the fixation task (*Figure 1d*, red lines), and we interpret these response increases as reflecting top-down modulation. In some cases, the top-down modulation is even larger than the modulation achieved by manipulation of the stimulus. For example, the VWFA response to 3%-contrast words during the one-back task exceeds the response to 100%-contrast words. Note that the task effects cannot be explained simply by differences in spatial attention: the one-back task produces substantially larger responses than the categorization task despite the fact that both tasks require the locus of spatial attention to be on the stimulus. Task effects in lower-level areas exist but are smaller in size (*Figure 1—figure supplement 1*).

A potential explanation of the top-down modulation is differences in task difficulty (*Ress et al., 2000*). For example, it is presumably more difficult to perceive low-contrast stimuli than high-

contrast stimuli, and this may explain why there is a large response enhancement for low- but not high-contrast stimuli (see VWFA contrast-response function for word stimuli in *Figure 1d*). Later in this paper, we provide a computational mechanism that could underlie the psychological concept of task difficulty.

In summary, our measurements indicate that VTC responses cannot be interpreted without specifying the cognitive state of the observer. A complete model of the computations performed by VWFA and FFA must consider the cognitive task in addition to stimulus properties.

## Model of bottom-up computations in VTC

Before addressing the influence of top-down factors, we first develop a model of bottom-up responses in VWFA and FFA. Although the field has long understood that stimulus category is a good predictor of evoked responses (*Kanwisher et al., 1997*; *Kriegeskorte et al., 2008*; *Grill-Spector and Weiner, 2014*), we do not yet have a computational explanation of this phenomenon. In other words, although we are able to use our own visual systems to assign a label such as 'word' or 'face' to describe the data, we have not yet identified the operations that enable our visual systems to derive these labels in the first place. An additional limitation of our conceptual understanding is that it fails to account for the sensitivity of VWFA and FFA to low-level image properties. We therefore ask: Is it possible to develop a quantitative characterization of the bottom-up computations that can reproduce observed stimulus selectivity in human VTC?

Extending an existing computational model of fMRI responses in the visual system (*Kay et al., 2008*, *2013b*, *2013c*), we conceive of a model involving two stages of image computations (*Figure 2a*). The first stage consists of a set of local oriented filters, akin to what has been used to model physiological responses in V1 (*Jones and Palmer, 1987*; *Carandini et al., 2005*). The second stage consists of a normalized dot product applied to the outputs of the first stage. This dot product computes how well a given stimulus matches a category template (for example, a word template for VWFA, a face template for FFA). We construct category templates directly from the stimulus set used in the experiment; in a later section we explore how well this approach generalizes. The present model, termed the *Template model*, is almost certainly an oversimplification of the complex non-linear processing performed in VTC. Nevertheless, the model is theoretically motivated, consistent with hierarchical theories of visual processing (*Fukushima, 1980*; *Heeger et al., 1996*; *Serre et al., 2007*; *DiCarlo et al., 2012*; *Rolls, 2012*), and provides a useful starting point for characterizing the computations that underlie word- and face-selectivity. Moreover, unlike recently popular deep neural network models that also involve hierarchical processing (*Khaligh-Razavi and Kriegeskorte, 2014*; *Yamins et al., 2014*; *Güçlü and van Gerven, 2015*), the model we propose is parsimonious with only three free parameters, and is therefore straightforward to fit and interpret (see *Figure 2a* and Materials and methods).

Applying the Template model to responses measured during the fixation task, we find that the Template model accurately predicts a large amount of variance in the responses of VWFA and FFA (*Figure 2b*). The model outperforms a phenomenological model, termed the *Category model*, that posits that perceived stimulus category is sufficient to predict the response of category-selective regions. Notably, the Template model is able to predict the response to non-preferred stimulus categories in each ROI. This suggests that responses to non-preferred stimuli are meaningful and the result of a well-defined computation performed by the visual system (*Haxby et al., 2001*). The model also outperforms simplified versions of the Template model that include only one of the two processing stages, as well as versions of the Template model in which the category template lacks tuning (non-selective template), is equally weighted between words and faces (mixed template), or is constructed randomly (random template) (*Figure 2c*).

The experiment we have conducted explores a limited range of stimuli. To further assess how well the Template model generalizes, we collected an additional dataset that includes 92 images taken from a previous study of object representation (*Kriegeskorte et al., 2008*). In its original instantiation (*Figure 2a*), the Template model uses category templates tailored to the stimuli in the main experiment, and we find that this instantiation of the Template model does not generalize well to the larger stimulus set (*Figure 2—figure supplement 1b*). However, by implementing a simple model extension in which we use a data-driven approach to estimate category templates, we find that the Template model achieves a reasonable level of accuracy on the new stimulus set (*Figure 2— figure supplement 1d*). This finding validates the basic architecture of the Template model,

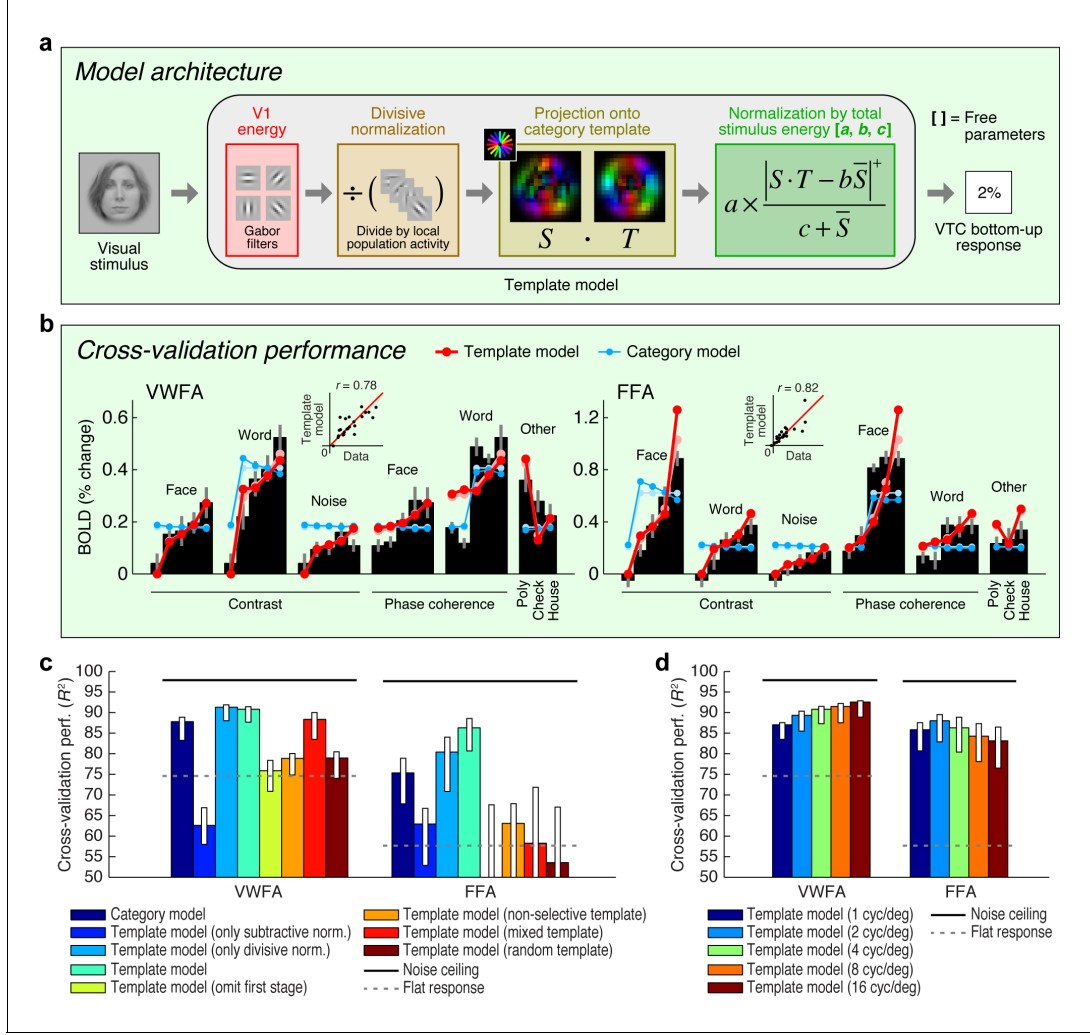

**Figure 2.** Model of bottom-up computations in VTC. (**a**) *Model architecture*. The predicted response of the Template model is given by a series of image computations (see Materials and methods). (**b**) *Cross-validation performance*. Black bars indicate bottom-up stimulus-driven responses measured during the fixation task, dark lines and dark dots indicate model predictions (leave-one-stimulus-out cross-validation), and light lines and light dots indicate model fits (no cross-validation). Scatter plots in the inset compare model predictions against the data. The Template model is compared to the Category model which simply predicts a fixed response level for stimuli from the preferred stimulus category and a different response level for all other stimuli (the slight decrease in response as a function of contrast is a result of the cross-validation process). (**c**) *Comparison of performance against control models*. Bars indicate leave-one-stimulus-out cross-validation performance. Error bars indicate 68% CIs, obtained by bootstrapping (resampling subjects with replacement). Solid horizontal lines indicate the noise ceiling, that is, the maximum possible performance given measurement variability in the data. Dotted horizontal lines indicate the cross-validation performance of a model that predicts the same response level for each data point (this corresponds to $R^2 = 0$ in the conventional definition of $R^2$ where variance is computed relative to the mean). The performance of the Template model degrades if the second stage of nonlinearities is omitted (Template model (only subtractive normalization)) or if the first stage of the model involving V1-like filtering is omitted (Template model (omit first stage)). The plot also shows that the precise configuration of the template is important for achieving high model performance (Template model (non-selective, mixed, random templates)). (**d**) *Performance as a function of spatial frequency tuning*. Here we manipulate the spatial frequency tuning of the filters in the Template model (while fixing spatial frequency bandwidth at one octave). The Template model uses a single set of filters at a spatial frequency tuning of 4 cycles/degree.

The following figure supplement is available for figure 2:

**Figure supplement 1.** Testing the Template model on a wide range of stimuli.

demonstrates how the Template model might be extended to account for increasingly large ranges of measurements, and provides a promising method to model response properties of other high-level visual regions not investigated here (for example, place- and limb-selective cortex).

The Template model advances us towards a computational understanding of VTC by demonstrating that BOLD responses in VTC can be predicted based on a template-matching operation on incoming visual inputs filtered by early visual cortex. The present results indicate that although high-level representations are not identical to low-level properties, they are built from, and fundamentally tied to, low-level properties through a series of linear and nonlinear operations. This conclusion is consistent with classic hierarchical theories of visual cortex (*Fukushima, 1980*; *Heeger et al., 1996*; *Serre et al., 2007*; *DiCarlo et al., 2012*; *Rolls, 2012*) and recent evidence that visual features may explain semantic representations found in high-level visual cortex (*Jozwik et al., 2016*). Our model can be viewed as a potential mechanism for how semantic tuning properties emerge in visual cortex (*Huth et al., 2012*). Our results indicate when studying high-level sensory representations in the brain, a precise characterization of the stimulus still matters.

## Top-down modulation acts as a stimulus-specific scaling

While the Template model explains bottom-up responses of VWFA and FFA as indexed by the fixation task, it does not explain why responses are higher in these areas during the categorization and one-back tasks (see *Figure 1d*). This is simply because the stimuli are identical across the three tasks and the response of the Template model, like that of many computational models of visual processing, is solely a function of the stimulus. Before we can design a model to capture the top-down effects, we must first understand exactly how top-down signals shape the VTC response.

By visualizing VTC responses as points in a multi-dimensional neural space with VWFA, FFA, and hV4 BOLD response amplitudes as the axes, we see that the responses to words and faces lie on specific manifolds, appearing as 'arms' that emanate from the origin (*Figure 3*). Importantly, we observe that the categorization and one-back tasks act as a scaling mechanism on the representation observed during the fixation task. The scaling mechanism moves the representation of each stimulus along the arms and away from the origin. Moreover, the amount of scaling is not constant across stimuli but is stimulus-specific, and this is most evident when considering the lowest contrast stimuli (*Figure 3*, black dots).

The visualization also shows that substantial responses to non-preferred categories are present in each ROI (for example, faces in VWFA, words in FFA) and that these responses are scaled during the stimulus-directed tasks. Thus, not only is information regarding non-preferred categories present in each ROI, but this information is actively modulated when subjects perform a perceptual task on those categories. These observations support the view that the brain uses a distributed strategy for perceptual processing and that category-selective regions are components of a more general network of regions that coordinate to extract visual information (*Haxby et al., 2001*; *Cox and Savoy, 2003*). An alternative scheme, more in line with a modular view of perceptual processing (*Kanwisher and Wojciulik, 2000*; *Baldauf and Desimone, 2014*), is area-specific enhancement, in which the representation of a stimulus is enhanced only in the region that is selective for that stimulus (for example, enhancement of words only in VWFA, enhancement of faces only in FFA). This scheme is not supported by our measurements (*Figures 3b and 3c*; formal model evaluation is performed in a later section). Rather, response scaling occurs even for non-preferred stimulus categories, and the amount of scaling varies as a function of stimulus properties such as image contrast.

A simple interpretation of the scaling effects is that they serve to increase signal-to-noise ratio in visually evoked responses in VTC (*Brouwer and Heeger, 2013*). For example, assuming that one use of the stimulus representation in VTC is to discriminate whether the presented stimulus is a word or face (or, more generally, identify the category of the stimulus [*DiCarlo et al., 2012*]), the scaling induced by the stimulus-directed tasks serves to increase the distance of neural responses from a linear decision boundary that separates words and faces (*Figure 3d*). Interestingly, the categorization and one-back tasks appear to act via the same scaling mechanism. The stronger scaling observed for the one-back task might be a consequence of increased amplitude or duration of neural activity. These results suggest that, at least for the perceptual tasks sampled here and the spatial scale of neural activity measured in this study, top-down cognitive processes do not impart additional tuning or selectivity but serve to amplify the selectivity that is already computed by visual cortex.

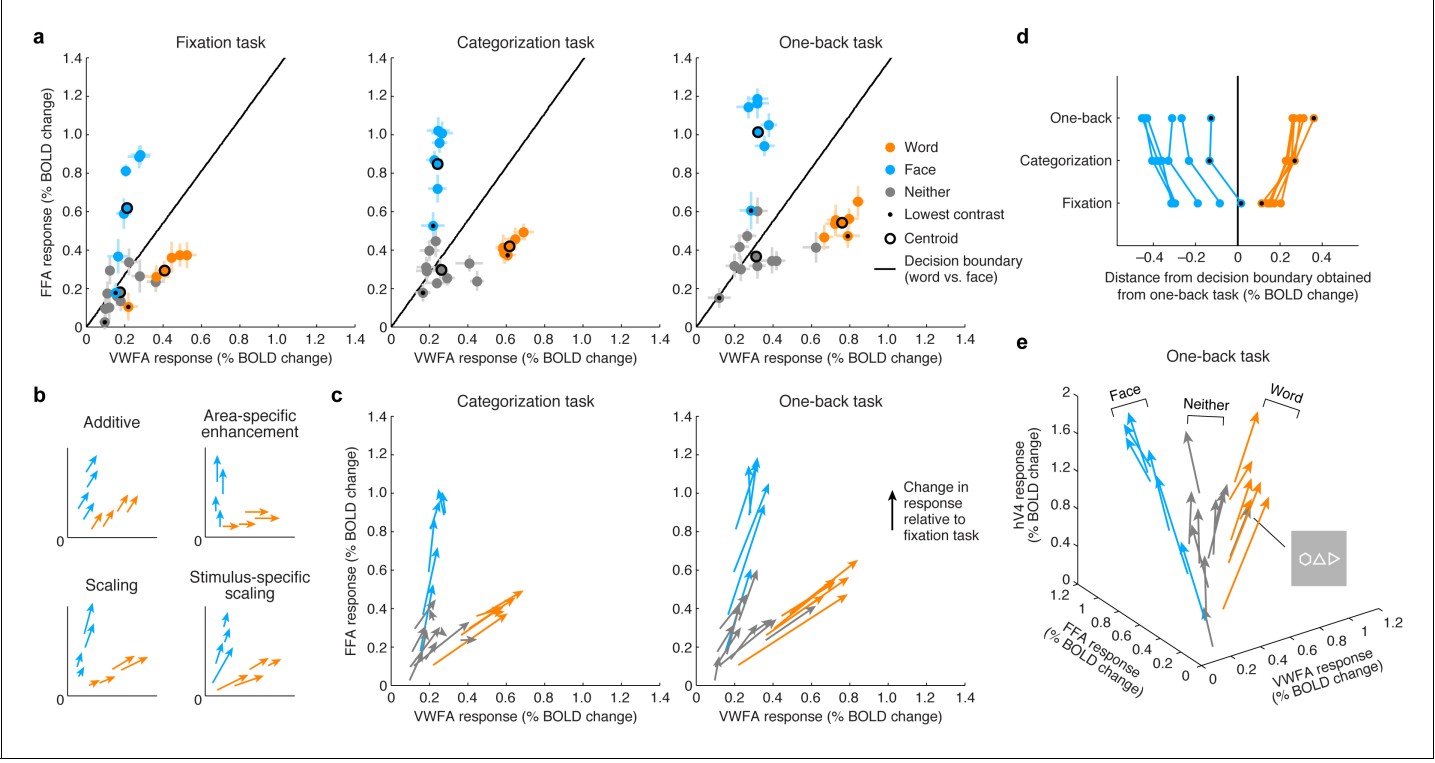

**Figure 3.** Top-down stimulus-specific scaling of VTC representation. (**a**) *Responses plotted in multi-dimensional neural space.* Each dot indicates ROI (VWFA, FFA) responses to a stimulus. In each plot, the black line indicates a linear decision boundary separating words and faces (nearest-centroid classifier, angular distance). (**b**) *Schematics of potential top-down mechanisms* (these models are formally evaluated in *Figure 5c*; see Materials and methods section 'IPS-scaling model' for details). (**c**) *Categorization and one-back tasks produce stimulus-specific scaling.* Arrows indicate the change in representation compared to the fixation task. (**d**) *Scaling improves readout.* Each data point indicates the signed Euclidean distance between the word-face decision boundary (as determined from the one-back task) and the neural response to a single stimulus. Lines join data points that correspond to the same stimulus. The scaling observed during the categorization and one-back tasks moves responses away from the decision boundary, thereby improving signal-to-noise ratio. (**e**) *Separation of other stimulus categories.* Including hV4 as a third dimension reveals that stimuli categorized as neither words nor faces manifest as a third 'arm' that emanates from the origin. Although not reported to be a word by the subjects, the polygon stimulus behaves similarly to word stimuli.

## IPS is the source of top-down modulation to VTC

To design a plausible model that can predict top-down effects, we next turn to identifying the neural circuitry that generates task modulations in VTC. There are two candidate mechanisms. The first is that sensitivity to task is locally generated from the neuronal architecture of VTC itself. We explore an alternative hypothesis whereby top-down modulation is induced by input from another brain region that is sensitive to task demands. To identify this region, we perform a connectivity analysis in which we first subtract the bottom-up signal in VTC, as given by responses measured during the fixation task, from responses measured during the categorization and one-back tasks. We then correlate these residuals, which isolate the top-down signal, against the responses of every cortical location.

Applying this connectivity analysis to our data, we find that responses in the intraparietal sulcus (IPS) predict the top-down enhancement of VTC responses (*Figure 4b*) better than responses in any other region of cortex. As a control, if we omit the subtraction step and simply correlate raw VTC responses with the responses of different cortical locations, we find that the correlation is instead strongest with a range of areas spanning occipital cortex (*Figure 4a*). This indicates that the VTC response is a mixture of bottom-up and top-down effects and that the top-down influence from the IPS becomes clear only when bottom-up effects are removed. Comparing our results to a publicly available atlas (*Wang et al., 2015*), we estimate that the source of top-down modulation is localized to the IPS-0 and IPS-1 subdivisions of the IPS (see also *Figure 4—figure supplement 1*).

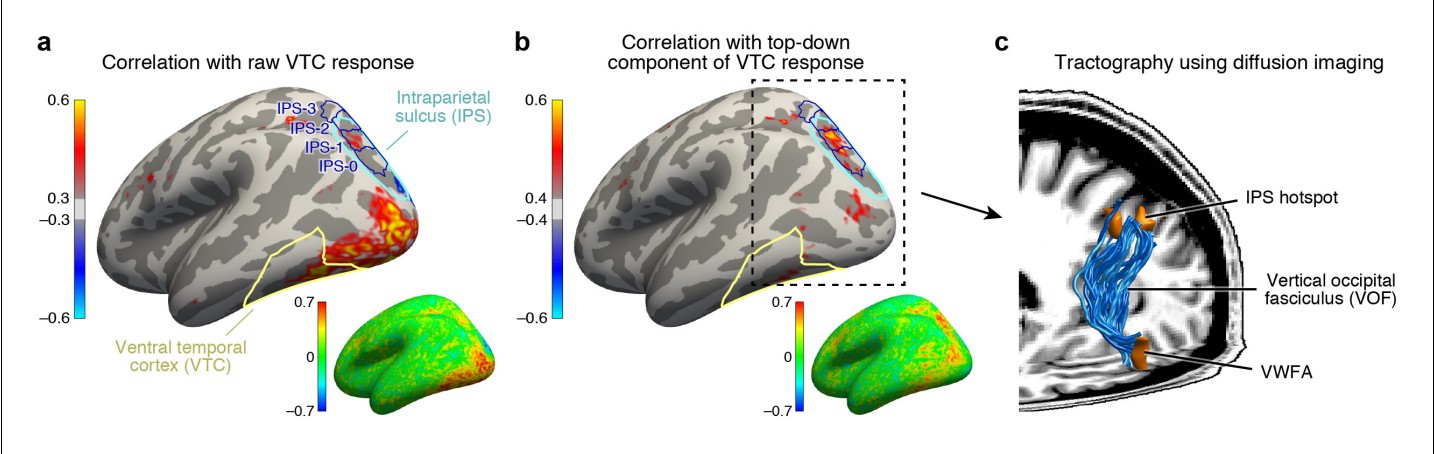

**Figure 4.** IPS is the source of top-down modulation to VTC. (**a**) *Correlation with raw VTC response*. This map depicts the correlation between the VTC response observed during the categorization and one-back tasks with the response at each cortical location (inset shows an unsmoothed and unthresholded map). Positive correlations are broadly distributed across occipital cortex. Results are shown for subjects with whole-brain coverage (*n* = 3); results for other subjects with partial-brain coverage (*n* = 6) are shown in *Figure 4—figure supplement 1*. (**b**) *Correlation with top-down component of VTC response*. After removing bottom-up responses (fixation task), the correlation is spatially localized to a hotspot in IPS-0/1. (**c**) *Tractography using diffusion MRI*. We find that the vertical occipital fasciculus (*Yeatman et al., 2014*) connects VWFA and FFA to the IPS hotspot in each subject for which diffusion data were collected (*n* = 8) (rendering shows a representative subject).

The following figure supplement is available for figure 4:

**Figure supplement 1.** Maps of top-down connectivity to VTC.

Previous research has identified IPS as playing a key role in controlling spatial attention (*Saalmann et al., 2007*; *Lauritzen et al., 2009*). Our results extend these findings by showing that, despite the fact that spatial attention is always directed towards the foveal stimulus during the categorization and one-back tasks, the amount of modulation from the IPS is flexible and varies depending on properties of the stimulus and demands of the task. For example, during the categorization task, the observed enhancement for low-contrast stimuli is much larger than that for high-contrast stimuli. This mechanism could explain the finding that difficult tasks enhance visual responses (*Ress et al., 2000*).

The direct influence of IPS on neural responses in VTC is consistent with anatomical measurements demonstrating the existence of a large white-matter pathway connecting dorsal and ventral visual cortex, called the vertical occipital fasciculus (VOF) (*Yeatman et al., 2013*, *2014*; *Takemura et al., 2016*). Using diffusion-weighted MRI and tractography (data acquired in 8 of 9 subjects), we show that the VOF specifically connects the VWFA and FFA with the functionally identified peak region in the IPS (*Figure 4c*). The VWFA falls within the ventral terminations of the VOF for seven subjects and, for the eighth, the VWFA is 2.7 mm anterior to the VOF, well within the margin of error for tractography (*Jeurissen et al., 2011*). The FFA falls within the ventral terminations of the VOF for all eight subjects. These results provide an elegant example of how anatomy subserves function, and sets the stage for a circuit-level computational model that, guided by anatomical constraints, characterizes the computations that emerge from interactions between multiple brain regions.

## Model of top-down computations in VTC

The previous two sections provide critical insights that set the stage for building a quantitative model that predicts top-down effects in VTC. Building upon the observation that top-down modulation acts as a scaling mechanism on responses in VWFA and FFA (see *Figure 3*) and the observation that top-down effects are correlated with the IPS signal (see *Figure 4*), we propose that the magnitude of the IPS response to a stimulus indicates the amount of top-down scaling that is applied to bottom-up sensory responses in VTC (*Figure 5a*). We implement this model, termed the *IPS-scaling*

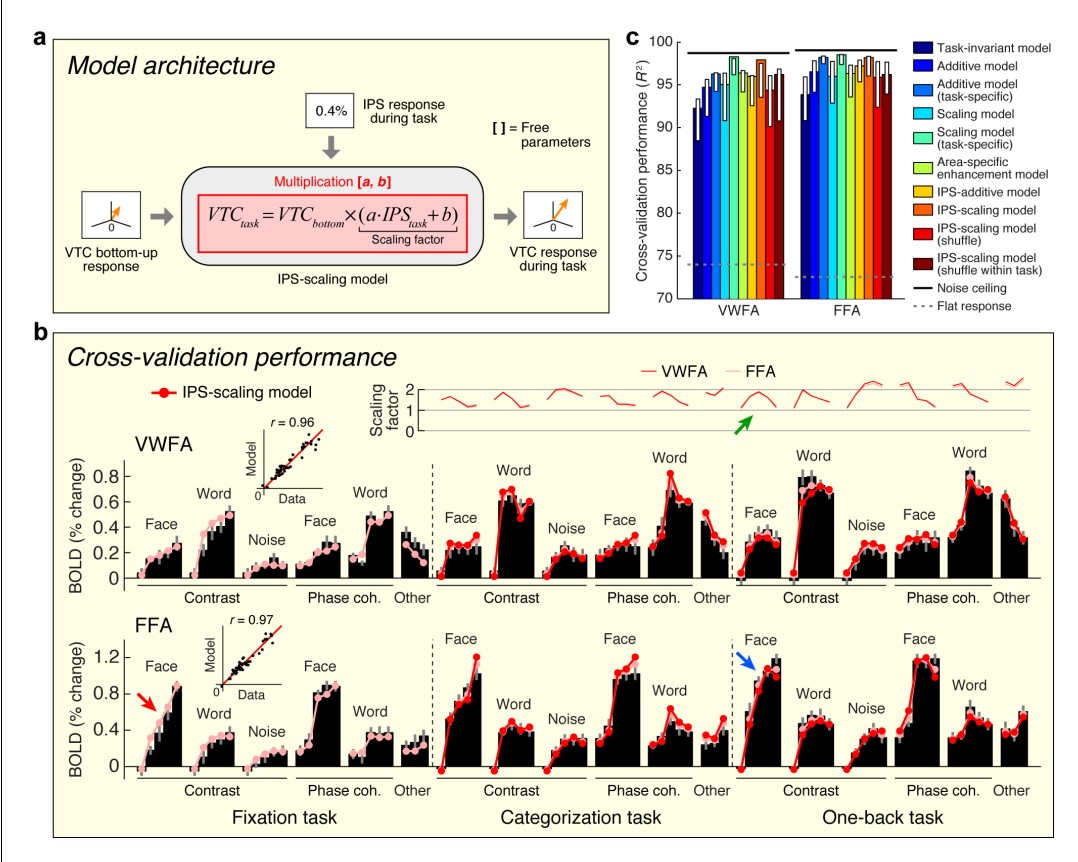

**Figure 5.** Model of top-down computations in VTC. (a) *Model architecture.* The predicted response during the stimulus-directed tasks (categorization task, one-back task) is given by scaling the bottom-up response, with the amount of scaling proportional to the IPS signal. (b) *Cross-validation performance.* Same format as *Figure 2b*. The arrows highlight an example of how the bottom-up response (red arrow) is multiplied by the IPS signal (green arrow) to produce the predicted response (blue arrow). (c) *Comparison of performance against alternative models.* Same format as *Figure 2c* (some error bars do not include the bar height; this is a consequence of the bootstrap procedure). Although the Additive and Scaling models perform well, note that these are *ad hoc*, phenomenological models. For instance, the Scaling model (task-specific) posits separate parameters for the amount of scaling under the categorization and one-back tasks. However, such a model does not explain *why* there is a different amount of scaling, whereas the IPS-scaling model provides such an explanation.

*model*, using response magnitudes extracted from a broad anatomical mask of the IPS. This strategy helps avoids the overfitting that might ensue from a more specific voxel-selection procedure tailored to the fine-scale and potentially idiosyncratic pattern of results from the connectivity analysis. For example, if we were to select the single cortical location in the IPS that best correlates with the top-down modulation of VTC, this would make voxel selection a critical part of the model and render the modeling analysis circular (*Kriegeskorte et al., 2009*). Nevertheless, the selection procedure is not completely independent, so the modeling results should not be viewed as providing independent evidence for the involvement of the IPS.

We find that the IPS-scaling model accurately characterizes the observed data (*Figure 5b*). For example, notice that the FFA response to faces increases gradually for each contrast increment during the fixation task (relatively unsaturated contrast-response function, red arrow). When subjects perform the one-back task, we observe a U-shaped contrast-response function in IPS (green arrow); multiplication of the two functions predicts a contrast-response function that is highly saturated and accurately matches the observed contrast-response function in FFA during the one-back task (blue arrow).

Importantly, the IPS-scaling model uses a single set of scale and offset parameters on the IPS response and accurately predicts scaling of VTC responses across the categorization and one-back tasks (*Figure 5b*, top plot). This finding suggests that the scaling of VTC by IPS is a general

mechanism supporting perception and is independent of the specific cognitive task performed by the observer. Furthermore, the scale and offset parameters that are estimated from the data show that when IPS exhibits close to zero evoked activity (for example, FACE at 100%-contrast; see *Figure 1—figure supplement 1*), the corresponding scaling factor is close to one. This has a sensible interpretation: when IPS is inactive, we observe only the bottom-up response in VTC and no top-down modulation.

We assessed the cross-validation performance of the IPS-scaling model in comparison to several alternative models of top-down modulation (including those schematized earlier in *Figure 3b*). In line with earlier observations (*Figure 3b and c*), we find that a model positing enhancement for only the preferred stimulus category of each area (Area-specific enhancement model) does not optimally describe the data. We find that a phenomenological scaling model (Scaling model (task-specific)) outperforms a phenomenological additive model (Additive model (task-specific)), confirming earlier observations that the top-down modulation is a scaling effect (*Figure 5c*). This conclusion is further supported by the higher performance observed when the IPS interacts with VTC multiplicatively (IPS-scaling model) compared to when it interacts additively (IPS-additive model). Finally, we find that the performance of the IPS-scaling model degrades if the IPS input into the model is shuffled across conditions (IPS-scaling model (shuffle, shuffle within task)), confirming that top-down modulation from the IPS is dependent on the stimulus and task.

Is the IPS the only region that induces top-down modulation of VTC? Inspection of the connectivity results (see *Figure 4b*) reveals that the top-down residuals in VTC are correlated, to a lesser extent, with the responses of other regions. These weaker correlations might be incidental, or might capture other important signals. Given that the IPS-scaling model accounts for nearly all of the variance induced by top-down modulation of VTC (see *Figure 5b and c*), we suggest that it is sufficient to consider only the IPS for the current set of measurements. However, future measurements that employ new stimulus manipulations and other cognitive tasks may reveal the role of a more extensive brain network. The IPS-scaling model can be extended to account for new measurements by systematically parameterizing the connectivity with additional brain regions. For example, some models of reading posit that language-related regions can directly influence the VWFA (*Twomey et al., 2011*), suggesting that to account for measurements made during a more naturalistic reading task, it may be necessary to include Broca's area in the model.

## Model of perceptual decision-making in IPS

Although informative, the finding that IPS provides top-down stimulus-specific scaling of VTC is an incomplete explanation, as the burden of explaining the top-down effects is simply shifted to the IPS. We are thus left wondering: is it possible to explain the response profile of the IPS? In particular, can we explain why the IPS is more active for certain stimuli compared to others? Answering these questions will provide a critical link between cognitive state and IPS activity.

Inspired by previous research on perceptual decision-making (*Shadlen and Newsome, 2001*; *Heekeren et al., 2004*; *Gold and Shadlen, 2007*; *Kayser et al., 2010a*), we implement a *Drift diffusion model* that attempts to account for IPS responses measured during the categorization task (*Figure 6a*). The model uses VTC responses during the fixation task as a measure of sensory evidence, and posits that the IPS accumulates evidence from VTC over time and exhibits an activity level that is monotonically related to accumulation time. For example, when VTC responses are small, as is the case for low-contrast stimuli, sensory evidence for stimulus category is weak, leading to long accumulation times (indexed by measurements of reaction time during the experiment), and large IPS responses.

Our implementation of the Drift diffusion model involves two steps. First, we use VTC responses during the fixation task (reflecting sensory evidence) to predict reaction times measured in the categorization task. The quality of the predictions is quite high (*Figure 6b*, left). Second, we apply a simple monotonic function to the reaction times measured during the categorization task to predict the level of response in the IPS (see Materials and methods). The rationale is that neural activity in IPS is expected to be sustained over the duration of the decision-making process (*Shadlen and Newsome, 2001*), and so the total amount of neural activity integrated over time should be larger for longer decisions. Assuming that the BOLD signal reflects convolution of a sluggish hemodynamic response function and fine-scale neural activity dynamics, small differences in the duration of neural activity (for example, between 0 and 2 s) are expected to manifest in differences in BOLD amplitudes

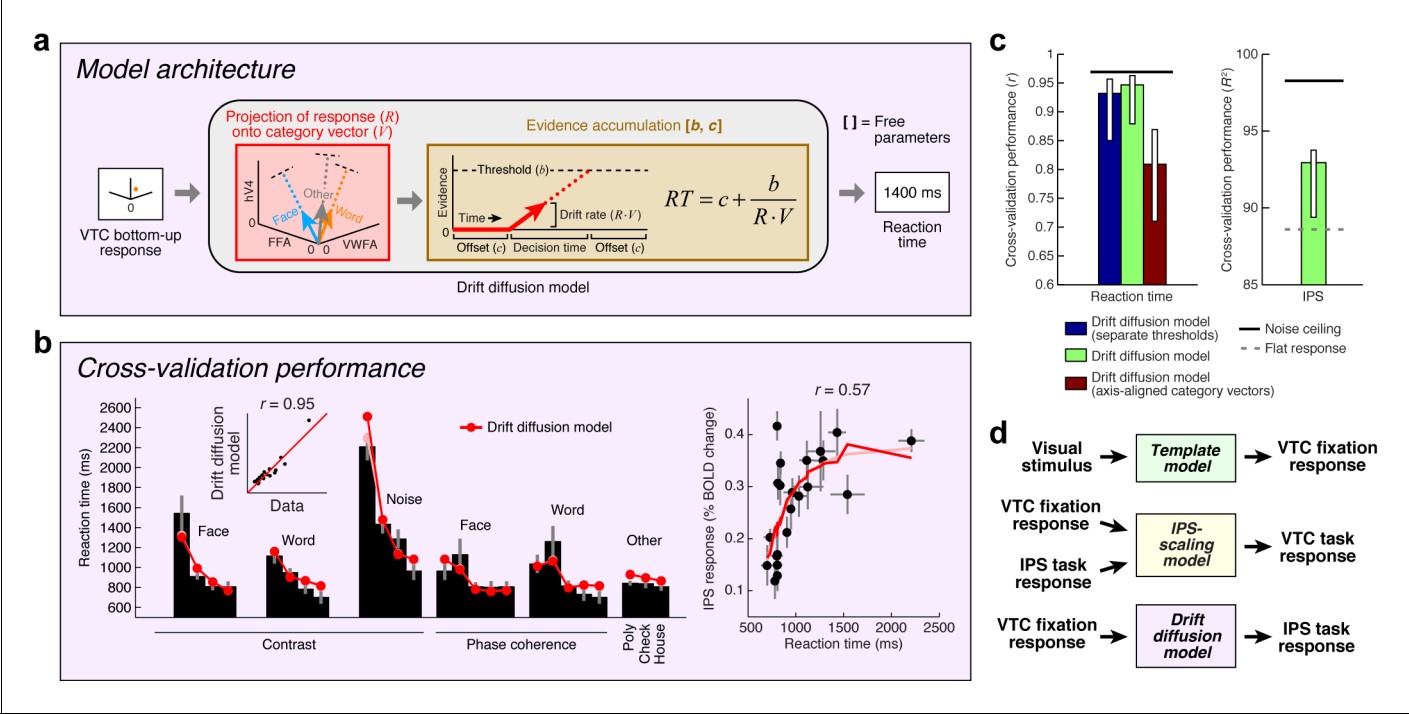

**Figure 6.** Model of perceptual decision-making in IPS. (a) *Model architecture*. We implement a model that links the stimulus representation in VTC to a decision-making process occurring in IPS. The model first uses the bottom-up VTC response as a measure of sensory evidence and predicts reaction times in the categorization task. The model then predicts the IPS response as a monotonically increasing function of reaction time. Note that this model does not involve stochasticity in the evidence-accumulation process, and is therefore a simplified version of the classic drift diffusion model (*Ratcliff, 1978*). (b) *Cross-validation performance*. Same format as *Figure 2b* (except that reaction times are modeled in the left plot). (c) *Comparison of performance against control models*. The performance of the Drift diffusion model does not degrade substantially if a single threshold is used, thus justifying this simplification. Performance degrades if axis-aligned category vectors are used, supporting the assertion that responses of multiple VTC regions are used by subjects in deciding image category. (d) *Overall model architecture*. This schematic summarizes all components of our computational model (*Figures 2a, 5a* and *6a*). Bottom-up visual information is encoded in the VTC fixation response (green box; Template model), fixation responses are routed to the IPS for evidence accumulation (purple box; Drift diffusion model), and then feedback from the IPS to VTC causes top-down modulation during the categorization and one-back tasks (yellow box; IPS-scaling model).

(*Kayser et al., 2010a*) and only minimally in the shapes of BOLD timecourses. The cross-validated predictions of our proposed model explain substantial variance in IPS (*Figure 6b*, right).

It is possible to offer a psychological explanation of IPS activity as reflecting task difficulty—for example, we can posit that IPS activity is enhanced for low-contrast stimuli because the observer works harder to perceive these stimuli. The value of the model we have proposed is that it provides a quantitative and formal explanation of the computations that underlie 'difficulty'. According to the model, categorization of low-contrast stimuli is difficult because the IPS computations required to perform the task involve longer accumulation time, and this is reflected in the fact that IPS response magnitudes increase monotonically with reaction time. Thus, our model performs several critical functions: it relates the cognitive task performed by the subject to IPS activity, proposes a computational explanation of task difficulty, and posits that top-down modulation of VTC by IPS is a direct consequence of fulfilling task demands. We have substantiated this hypothesis for the categorization task and suggest that this will serve as a foundation for modeling more complex cognitive tasks such as one-back.

## Discussion

In summary, we have measured and modeled how bottom-up and top-down factors shape responses in word- and face-selective cortex. A template operation on low-level visual properties

generates a bottom-up stimulus representation, while top-down modulation from the IPS scales this representation in the service of the behavioral goals of the observer. We develop a computational approach that posits explicit models of the information processing performed by a network of interacting sensory and cognitive regions of the brain and validate this model on experimental data. We make publicly available data and open-source software code implementing the model at *Kay, 2017* (with a copy archived at https://github.com/elifesciences-publications/vtcipsmodel).

The model we propose is valuable because it integrates and explains a range of different stimulus and task manipulations that affect responses in VWFA, FFA, and IPS. Response properties in these regions can now be interpreted using a series of simple, well-defined computations that can be applied to arbitrary images. However, it is also important to recognize the limitations of the model. First, we have tested the model on only a limited range of stimuli and cognitive tasks. Second, the accuracy with which the model accounts for the data is reasonable but by no means perfect. For instance, the Template model does not capture the step-like response profile of VTC as phase coherence is varied (see *Figure 2b*), and the accuracy with which the model accounts for a wide range of stimuli that includes faces, animals, and objects, is moderate at best (see *Figure 2—figure supplement 1*). Third, we have thus far characterized VTC and IPS responses at only a coarse spatio-temporal scale (that is, BOLD responses averaged over specific regions-of-interest). Given these limitations, the present work constitutes a first step towards the goal of developing a comprehensive computational model of human high-level visual cortex. We have provided data and code so that other researchers can build on our approach, for example, by testing the generalizability of the model to other stimuli and tasks, extending and improving the model, and comparing the model against alternative models.

The fact that cognitive factors substantially affect stimulus representation in visual cortex highlights the importance of tightly controlling and manipulating cognitive state when investigating stimulus selectivity. In the present measurements, the most striking example comes from stimulus contrast. When subjects perform the fixation task, the contrast-response function (CRF) in VWFA is monotonically increasing, whereas during the one-back task, the CRF flips in sign and is monotonically decreasing (see *Figure 1d*). This effect (similar to what is reported in *Murray and He, 2006*) is puzzling if we interpret the CRFs as indicating sensitivity to the stimulus contrast, but is sensible if we interpret the CRFs as instead reflecting the interaction of stimulus properties and cognitive processes. The influence of cognition on visual responses forces us to reconsider studies that report unexpected tuning properties in VTC and IPS, such as tuning to linguistic properties of text in VWFA (*Vinckier et al., 2007*) and object selectivity in parietal cortex (*Sereno and Maunsell, 1998*). In experiments that do not tightly control the cognitive processes executed by the observer, it is impossible to distinguish sensory effects from cognitive effects. Our quantitative model of VTC-IPS interactions provides a principled baseline on which to re-interpret past findings, design follow-up experiments, and guide data analysis.

There is a large body of literature on characterizing and modeling the effect of spatial attention on contrast-response functions (*Luck et al., 1997*; *Boynton, 2009*; *Reynolds and Heeger, 2009*; *Itthipuripat et al., 2014*). Although several models have been proposed, none of these straightforwardly account for the present set of measurements. The *response-gain* model (*McAdams and Maunsell, 1999*) posits that attention causes a multiplicative scaling of contrast-response functions. Our observations are consistent with the general notion of response scaling (see *Figure 3*), but importantly, we find that the amount of scaling differs for different stimuli. Whereas the response-gain model implies that scaling is constant and therefore contrast-response functions should grow steeper during the stimulus-directed tasks, we find the opposite (see *Figure 1d*). The *contrast-gain* model (*Reynolds et al., 2000*) posits that attention causes a leftward shift of contrast-response functions (as if contrast were increased). This model does not account for our measurements, since the stimulus-directed tasks can generate responses to low-contrast stimuli that are larger than responses to high-contrast stimuli (see *Figure 1d*). Finally, the *additive-shift* model (*Buracas and Boynton, 2007*) posits that attention causes an additive increment to contrast-response functions; we find that our observations are better explained by a scaling, not additive, mechanism (see *Figures 3* and *5c*). Thus, the effects we report are novel, and previous models of attention cannot explain these effects. Furthermore, by investigating responses to a wide range of stimuli (including manipulations of not only contrast but also phase coherence and stimulus category) and by characterizing the

source of attentional signals, our work develops a more comprehensive picture of information processing in the visual system.

There are a number of research questions that remain unresolved. First, our connectivity analysis and modeling of VTC-IPS interactions are based on the correlation of BOLD responses and do not provide information regarding the directionality, or timing, of neural interactions. In other words, our correlational results do not, in and of themselves, prove that the IPS causes VTC modulation; rather, we are imposing an interpretation of the results in the context of a computational model. We note that our interpretation is in line with previous work on perceptual decision-making showing top-down influence (Granger causality) of IPS on visual cortex (*Kayser et al., 2010b*). Our working hypothesis is that sensory information arrives at VTC (as indexed by fixation responses), these signals are routed to IPS for evidence accumulation, and then feedback from the IPS modulates the VTC response (as indexed by categorization and one-back responses). Temporally resolved measurements of neural activity (for example, EEG, MEG, ECoG) will be necessary to test this hypothesis. Second, the scaling of BOLD response amplitudes by IPS is consistent with at least two potential mechanisms at neural level: the IPS may be inducing a scaling on neural activity in VTC or, alternatively, a sustainment of neural activity in VTC. Some support for the latter comes from a recent study demonstrating that ECoG responses in FFA exhibit sustained activity that is linked to long reaction times in a face gender discrimination task (*Ghuman et al., 2014*). Finally, IPS is part of larger brain networks involved in attention (*Corbetta and Shulman, 2002*) and decision-making (*Gold and Shadlen, 2007*), and identifying the computational roles of other regions in these networks is necessary for a comprehensive understanding of the neural mechanisms of perception.

## Materials and methods

### Subjects
Eleven subjects participated in this study. Two subjects were excluded due to inability to identify VWFA in one subject and low signal-to-noise ratio in another subject, leaving a total of nine usable subjects (age range 25–32; six males, three females). All subjects were healthy right-handed monolingual native-English speakers, had normal or corrected-to-normal visual acuity, and were naive to the purposes of the experiment. Informed written consent was obtained from all subjects, and the experimental protocol was approved by the Washington University in St. Louis Institutional Review Board. Each subject participated in 1–3 scanning sessions, over the course of which anatomical data (T1-weighted high-resolution anatomical volume, diffusion-weighted MRI data) and functional data (retinotopic mapping, functional localizer, main experiment) were collected.

### Visual stimuli
Stimuli were presented using an NEC NP-V260X projector. The projected image was focused onto a backprojection screen and subjects viewed this screen via a mirror mounted on the RF coil. The projector operated at a resolution of 1024 × 768 at 60 Hz, and the viewing distance was 340 cm. A Macintosh laptop controlled stimulus presentation using code based on the Psychophysics Toolbox (*Brainard, 1997*; *Pelli, 1997*). Approximate gamma correction was performed by taking the square root of pixel intensity values before stimulus presentation. Behavioral responses were recorded using a button box.

The experiment consisted of 22 types of stimuli. All stimuli were small grayscale images (approximately 2° × 2°) presented at fixation. Each stimulus type consisted of 10 distinct images (for example, 10 different faces for a face stimulus), and a subset of these images were presented on each given trial.

#### Face
This stimulus consisted of a face pictured from a frontal viewpoint. Ten distinct faces were prepared. Faces were masked using a circle with diameter 2°. The outer 0.25° of the mask was smoothly ramped using a cosine function.

### Word

This stimulus consisted of a 5-letter word. Ten distinct words were prepared. Letters were white on a gray background, generated using the Helvetica font, and occupied a rectangular region measuring 3.15° × 1.05°.

### Phase coherence

These stimuli consisted of the FACE and WORD stimuli prepared at four phase-coherence levels, 0%, 25%, 50%, and 75% (8 stimuli total). To achieve this, for each of the ten images from each stimulus type, the portion of the image within a fixed region (FACE: 2° × 2° square; WORD: 3.15° × 1.05° rectangle) was extracted, and its phase spectrum was blended, to different degrees, with a randomly generated phase spectrum. For example, 25% coherence indicates that the phase of each Fourier component was set to a value that lies at 75% of the angular distance from the original phase to the phase in the randomly generated spectrum.

### Noise

This stimulus is the same as the FACE stimulus at 0% phase coherence. For brevity, we refer to this stimulus as NOISE.

### Contrast

These stimuli consisted of the FACE, WORD, and NOISE stimuli prepared at three contrast levels (nine stimuli total). The contrasts of the original stimuli were taken to be 100%, and different contrast levels were achieved by scaling pixel intensity values towards the background value. Contrast levels of 4%, 6%, and 10% were used for the FACE and NOISE stimuli, and contrast levels of 3%, 5%, and 8% were used for the WORD stimulus. This choice of contrast levels matches the average root-mean-square (RMS) contrast across stimulus types (for example, the average RMS contrast for the FACE stimulus at 4% contrast is approximately equal to the average RMS contrast for the WORD stimulus at 3% contrast). Note that a contrast level of 0% was achieved by estimating responses to blank trials (see *GLM analysis*).

### Polygon

This stimulus consisted of a string of three polygons (each chosen randomly from a set of polygons). Polygons were white, unfilled, on a gray background, and occupied a region similar in size to that of the WORD stimulus. Ten distinct strings were prepared.

### Checkerboard

This stimulus consisted of alternating black and white square checks. Ten checkerboards were prepared by varying check size from 0.03125° to 0.5° using ten equally spaced steps on a logarithmic scale. The *x*- and *y*-positions of each checkerboard were set randomly. Checkerboards were masked using a circle with diameter 2°.

### House

This stimulus consisted of a house pictured from a frontal viewpoint. Ten distinct houses were prepared. Houses were masked using a 2° × 2° square. The outer 0.25° of the mask was smoothly ramped using a cosine function.

## Experimental design and tasks

Stimuli were presented in 4 s trials, one stimulus per trial. In a trial, four images from a given stimulus type (for example, FACE, 10% contrast) were presented sequentially using an 800 ms ON, 200 ms OFF duty cycle. To generate the sequence of four images, we first randomly selected four distinct images out of the ten images associated with the stimulus type. Then, for certain trials (details below), we modified the sequence to include a repetition by randomly selecting one of the images (excluding the first) and replacing that image with the previous image. Throughout stimulus presentation, a small dot (0.12° × 0.12°) was present at the center of the display. The dot switched to a

new randomly selected color every 600 ms using a set of six possible colors: magenta, red, yellow, green, cyan, and blue.

In the experiment, two of the stimuli were duplicated (FACE and WORD), yielding a total of 24 stimulus conditions. Data corresponding to these duplicate stimuli are not used in this paper. Each run began and ended with a 16 s baseline period in which no stimuli were presented. During a run, each of the 24 stimulus conditions was presented three times. Six blank trials (no stimulus) were also included. The order of stimulus and blank trials was random, subject to the constraints that blank trials could not occur first nor last, blank trials could not occur consecutively, and no stimulus condition could occur consecutively. During the baseline periods and blank trials, the small central dot was still present. A randomly selected two of the three trials associated with each stimulus condition were modified to include an image repetition (as described previously). Each run lasted 344 s (5.7 min).

For each run, subjects were instructed to maintain fixation on the central dot while performing one of three tasks. In the *fixation task*, subjects were instructed to press a button whenever the central dot turned red, and were additionally reminded to not confuse the red and magenta colors. In the *categorization task*, subjects were instructed to report for each stimulus trial whether they perceived a word, a face, or neither ('other'). Responses were made using three different buttons, and subjects were reminded to make only one response for each 4 s trial. Note that it is possible that responses are made prior to the completion of the four images that comprise a trial. In the *one-back task*, subjects were instructed to press a button whenever an image was repeated twice in a row, and were informed that repetitions occurred only within stimulus trials and not across trials. Subjects were warned that although some stimuli are faint (low contrast), they should still try their best to perform the categorization and one-back tasks. Subjects were also informed that some trials are blank trials and that responses were not expected on these trials. Subjects were familiarized with the stimuli and tasks before the actual experiment was conducted.

Subjects performed each of the three tasks four times during the course of the experiment, yielding a total of 3 tasks × 4 runs = 12 runs. The physical stimulus sequence (including the temporal ordering of stimulus images and dot colors) was held constant across tasks. This was accomplished by generating four distinct stimulus sequences and cycling through the sequences and tasks. Specifically, the order of stimulus sequences was ABCD ABCD ABCD, where each letter corresponds to a distinct sequence, and the order of tasks was XYZ XYZ XYZ XYZ, where each letter corresponds to a distinct task. The order of tasks was counterbalanced across subjects. Each stimulus and task combination (for example, CHECKERBOARD during one-back task) occurred a total of 3 trials × 4 runs = 12 times over the course of the experiment.

## MRI data acquisition

MRI data were collected at the Neuroimaging Laboratory at the Washington University in St. Louis School of Medicine using a modified 3T Siemens Skyra scanner and a 32-channel RF coil. For functional data, 28 oblique slices covering occipitotemporal cortex were defined: slice thickness 2.5 mm, slice gap 0 mm, field-of-view 200 mm × 200 mm, phase-encode direction anterior-posterior. A T2*-weighted, single-shot, gradient-echo EPI sequence was used: matrix size 80 × 80, TR 2 s, TE 30 ms, flip angle 77°, nominal spatial resolution 2.5 mm × 2.5 mm × 2.5 mm. Fieldmaps were acquired for post-hoc correction of EPI spatial distortion. To achieve comprehensive coverage for localization of top-down effects, a whole-brain version of the protocol involving 58 slices and a multiband (*Feinberg et al., 2010*) factor of 2 was used in three of the nine subjects. In addition to functional data, T1-weighted anatomical data (MPRAGE sequence, 0.8 mm resolution) and diffusion-weighted data (spin-echo EPI sequence, 2 mm resolution, 84 directions, *b*-values of 1500 and 3000) were acquired. The diffusion sequence was acquired twice, reversing the phase-encode direction, in order to compensate for spatial distortions. Diffusion data were not acquired for one subject due to time constraints.

## Behavioral analysis

Behavioral results for the categorization task are used in the present study. We analyzed both reaction times (RT) and category judgments. We defined RT as the time elapsed between the onset of the first of the four images in a given trial and the button press. Trials in which no buttons were pressed were ignored. For each subject, we summarized RTs by computing the median RT across

trials for each stimulus. To obtain group-averaged RTs, we added a constant to each subject's RTs in order to match the mean RT to the grand mean across subjects and then computed the mean and standard error across subjects (this normalization procedure compensates for additive offsets in RT across subjects). Category judgments were analyzed by calculating percentages of trials on which a given subject categorized a given stimulus into each of the three categories (word, face, other). Subjects were highly consistent in their judgments: for each stimulus, the most frequently reported category was the same across subjects and was reported more than 85% of the time. Category judgments obtained from the categorization task are used in the labeling and interpretation of experimental results (for example, *Figures 1* and *3*).

## Diffusion analysis

Subject motion was corrected by co-registering each volume to the average of the non-diffusion-weighted *b* = 0 images. Gradient directions were adjusted to account for the co-registration. From pairs of volumes acquired with reversed phase-encode directions, the susceptibility-induced off-resonance field was estimated using a method similar to that described in *Andersson et al. (2003)* as implemented in FSL (*Smith et al., 2004*). Eddy currents were corrected using FSL's *eddy* tool. The *b* = 3000 measurements were used to estimate fiber orientation distribution functions for each voxel using constrained spherical deconvolution as implemented in *mrtrix* (*Tournier et al., 2007*) (CSD, $l_{max}$ = 4), and fiber tracts were estimated using probabilistic tractography (500,000 fibers). For each subject, we identified the vertical occipital fasciculus (VOF) using a previously published algorithm (*Yeatman et al., 2014*), and then quantified the Euclidean distance from the VOF terminations to word- and face-selective regions in VTC and the task-related hotspot in the IPS.

## Pre-processing of anatomical and functional data

The T1-weighted anatomical volume acquired for each subject was processed using FreeSurfer (*Fischl, 2012*). The results were used to create a cortical surface reconstruction positioned halfway between the pial surface and the boundary between gray and white matter. We used the *fsaverage* surface from FreeSurfer to define anatomical ROIs (details below). These ROIs were transformed to native subject space by performing nearest-neighbor interpolation on the spherical surfaces created by FreeSurfer (these surfaces reflect folding-based alignment of individual subject surfaces to the *fsaverage* surface).

Functional data were pre-processed by performing slice time correction, fieldmap-based spatial undistortion, motion correction, and registration to the subject-native anatomical volume. The combined effects of distortion, motion, and registration were corrected using a single cubic interpolation of the slice time corrected volumes. Interpolations were performed directly at the vertices of the subject's cortical surface, thereby avoiding unnecessary interpolation and improving spatial resolution (*Kang et al., 2007*).

## GLM analysis

The pre-processed fMRI data were analyzed using GLMdenoise (*Kay et al., 2013a*) (http://kendrick-kay.net/GLMdenoise/), a data-driven denoising method that derives estimates of correlated noise from the data and incorporates these estimates as nuisance regressors in a general linear model (GLM) analysis of the data. For our experiment, we coded each stimulus and task combination as a separate condition and also included the blank trials, producing a total of (24 stimulus + 1 blank) × 3 tasks = 75 conditions. The response to blank trials was interpreted as the response to a 0%-contrast stimulus. Estimates of BOLD response amplitudes (beta weights) were converted to units of percent BOLD signal change by dividing amplitudes by the mean signal intensity observed at each vertex. To obtain ROI responses, beta weights were averaged across the vertices composing each ROI. Error bars (68% CIs) on beta weights were obtained by bootstrapping runs.

Group-averaged beta weights were calculated using a procedure that compensates for large intrinsic variation in percent BOLD change across subjects. First, the beta weights obtained for each subject in a given ROI were normalized to be a unit-length vector (for example, $\tilde{\mathbf{b}}_\mathbf{i} = \mathbf{b}_\mathbf{i}/\|\mathbf{b}_\mathbf{i}\|$ where $\mathbf{b}_\mathbf{i}$ indicates beta weights for the *i*th subject (1 x *n*), $\|\|$ indicates $L_2$-norm, and $\tilde{\mathbf{b}}_\mathbf{i}$ indicates normalized beta weights for the *i*th subject). Next, normalized beta weights were averaged across subjects, using bootstrapping to obtain error bars (68% CIs). Finally, the resulting group-averaged beta

weights were multiplied by a scalar such that the mean of the beta weights is equal to the mean of the original unnormalized beta weights obtained from all subjects. The motivation of this last step is to produce interpretable units of percent BOLD change instead of normalized units. Note that in some cases, beta weights are repeated for easier visualization (for example, in *Figure 1*, NOISE at 100% contrast is the same data point as FACE at 0% phase coherence). Group-averaged beta weights were used in computational modeling.

## Region-of-interest (ROI) definition

Visual field maps were defined using the population receptive field (pRF) technique applied to reti-notopic mapping data (*Dumoulin and Wandell, 2008*; *Kay et al., 2013b*). Subjects participated in 2–4 runs (300 s each) in which they viewed slowly moving apertures (bars, wedges, rings) filled with a colorful texture of objects, faces, and words placed on an achromatic pink-noise background. The aperture and texture were updated at 5 Hz, and blank periods were included in the design (*Dumoulin and Wandell, 2008*). A semi-transparent fixation grid was superimposed on top of the stimuli (*Schira et al., 2009*). Stimuli occupied a circular region with diameter 10° and the viewing distance was 251 cm. A small semi-transparent central dot (0.15° × 0.15°) was present throughout the experiment and changed color every 1–5 s. Subjects were instructed to maintain fixation on the dot and to press a button whenever its color changed. The time-series data from this experiment were modeled using the Compressive Spatial Summation model (*Kay et al., 2013b*) as implemented in analyzePRF (http://kendrickkay.net/analyzePRF/). Angle and eccentricity estimates provided by the model were then visualized on cortical surface reconstructions and used to define V1, V2, V3, and hV4 (*Brewer et al., 2005*). Due to the limited amount of pRF data acquired, there was insufficient signal-to-noise ratio to define visual field maps in parietal cortex.

Category-selective regions FFA and VWFA were defined using functional localizers (*Weiner and Grill-Spector, 2010*, *2011*). Subjects participated in two runs (336 s each) in which they viewed blocks of words, faces, abstract objects, and noise patterns. Each block lasted 16 s and consisted of 16 images presented at a rate of 1 Hz. The images differed from those in the main experiment. In each run, the four stimulus types were presented four times each in pseudorandom order, with occasional 16 s blank periods. A semi-transparent fixation grid was superimposed on top of the stimuli (*Schira et al., 2009*). Stimuli occupied a 4° × 4° square region, with the words, faces, and objects occupying the central 3° × 3° of this region. The viewing distance was 340 cm. Subjects were instructed to maintain central fixation and to press a button when the same image is presented twice in a row. The time-series data from this experiment were analyzed using a GLM to estimate the amplitude of the BOLD response to the four stimulus categories.

To define FFA and VWFA, responses to the four stimulus categories were visualized on cortical surface reconstructions. FFA and VWFA were defined based on stimulus selectivity, anatomical location, and topological relationship to retinotopic areas (*Weiner and Grill-Spector, 2010*; *Yeatman et al., 2013*; *Weiner et al., 2014*). We defined FFA as face-selective cortex (responses to faces greater than the average response to the other three categories) located on the fusiform gyrus. We included in the definition both the posterior fusiform gyrus (pFus-faces/FFA-1) and middle fusiform gyrus (mFus-faces/FFA-2) subdivisions of FFA (*Weiner et al., 2014*). We defined VWFA as word-selective cortex (responses to words greater than the average response to the other three categories) located in and around the left occipitotemporal sulcus. In some subjects, multiple word-selective patches were found, and all of these patches were included in the definition of VWFA.

Anatomically-defined ROIs were also created (see *Figure 4a and b*). Based on curvature values on the *fsaverage* surface, we created an anatomical mask of the IPS by selecting the posterior segment of the intraparietal sulcus (*Pitzalis et al., 2012*). Using the atlas of visual topographic organization provided by Wang et al. (*Wang et al., 2015*), we estimate that this IPS mask overlaps V3A, V3B, IPS-0, IPS-1, and IPS-2. The locations of IPS-0/1/2/3 from the atlas are shown in *Figure 4* and *Figure 4—figure supplement 1*. We also created an anatomical mask of VTC by computing the union of the *fusiform* and *inferiortemporal* parcels provided by the FreeSurfer Desikan-Killiany atlas (*Desikan et al., 2006*) and trimming the anterior extent of the result to include only visually responsive cortex. The VTC mask includes both FFA and VWFA as well as surrounding cortex.

In our data, we find that word-selective visual cortex in some subjects is confined to the left hemisphere, consistent with previous studies (*Yeatman et al., 2013*). Therefore, to ease interpretation, we restricted our analysis to VWFA, FFA, VTC, and IPS taken from the left hemisphere. In addition,

we restricted the definition of V1, V2, V3, hV4, VTC, and IPS to include only vertices exhibiting response amplitudes in the main experiment that are positive on average. This procedure excludes voxels with peripheral receptive fields which typically exhibit negative BOLD responses to centrally presented stimuli.

## Task-based functional connectivity

To identify the cortical region that generates top-down effects in VWFA and FFA, we performed a simple connectivity analysis. First, we averaged BOLD responses across our VTC mask, given that top-down effects appear broadly across VTC. Next, we identified the component of the VTC response that is of no interest, specifically, the bottom-up stimulus-driven response. Our estimate of this component is given by our measurement of VTC responses during the fixation task (22 stimuli + 1 blank = 23 values). We then subtracted the bottom-up component from the VTC response measured during the categorization task (23 values) and one-back task (23 values). This produced a set of residuals (46 values) that reflect the top-down effect in VTC. Finally, we correlated the residuals with the responses of each cortical location in our dataset during the categorization and one-back tasks (46 values). The cortical location that best correlates with the residuals is interpreted as a candidate region that supplies top-down modulation to VTC.

The results were visualized by averaging correlation values across subjects based on the *fsaverage* cortical alignment and plotting the results on the *fsaverage* surface. Results from the three subjects for which whole-brain fMRI data were acquired are shown in *Figure 4a and b*. Results from the remaining six subjects with limited fMRI coverage are provided in *Figure 4—figure supplement 1*. Note that the correlation-based analysis we have used is most suitable for connectivity effects that are additive in nature (for example, IPS providing additive enhancement to VTC). However, the modulation is more accurately characterized as a multiplicative, or scaling, effect (see *Figure 3b and c*). The advantage of correlation is that it is robust to noise and computationally efficient; we perform a more precise evaluation of different top-down mechanisms in the computational modeling section below.

There are three important differences between the connectivity analysis described here and conventional correlation-based resting-state functional connectivity (RSFC) (*Buckner et al., 2013*) and the psychophysiological interactions (PPI) technique (*O'Reilly et al., 2012*). One is that our connectivity is performed on data that have explicit manipulation of stimulus and task (unlike RSFC). Another is that we analyze the data explicitly in terms of information-processing operations performed by the brain (unlike PPI). In other words, functional connectivity is characterized, not as correlated signal fluctuations, but as a direct consequence of information-processing operations. A third difference is that our connectivity is performed on beta weights that pool across trials (*Rissman et al., 2004*), as opposed to raw BOLD time-series. This concentrates the analysis on brain responses that are reliably driven by the stimulus and task, and de-emphasizes trial-to-trial fluctuations in cognitive performance (*Donner et al., 2013*).

## Computational modeling

We developed a computational model to account for BOLD responses measured in VTC and IPS. The model is composed of three components, each of which addresses a different aspect of the data (*Figure 6d*). The first component (*Template model*) specifies how a given stimulus drives bottom-up VTC responses as measured during the fixation task; the second component (*IPS-scaling model*) specifies how top-down modulation from the IPS during the categorization and one-back tasks affects VTC responses; and the third component (*Drift-diffusion model*) specifies how accumulation of evidence from VTC predicts reaction times and IPS responses during the categorization task. Note that although the three model components could be yoked together (for example, the output from the Template model could serve as the input to the Drift-diffusion model), in our model implementations, we adopt the approach of isolating each model component so that the quality of each component can be assessed independently of the others.

For all three model components, computational modeling was performed using nonlinear least-squares optimization (MATLAB Optimization Toolbox). Leave-one-stimulus-out cross-validation was used to assess model accuracy (thus, we assess the ability of models to generalize to stimuli that the models have not been trained on). Note that the use of cross-validation enables fair comparison of

models that have different levels of flexibility (or, informally, different numbers of free parameters). This is because models that are overly complex will tend to fit noise in the training data and thereby generalize poorly to the testing data (*Hastie et al., 2001*).

Accuracy was quantified as the percentage of variance explained ($R^2$) between cross-validated predictions of the data (aggregated across cross-validation iterations) and the actual data. In the case of beta weights, variance was computed relative to 0% BOLD signal change (*Kay et al., 2013b*). In certain cases, accuracy is reported using Pearson's correlation (*r*); this metric assesses performance relative to the mean. To assess reliability of cross-validation results, model fitting and cross-validation were repeated for each bootstrap of the group-averaged data (resampling subjects with replacement). For benchmarks on cross-validation performance, we calculated noise ceilings using Monte Carlo simulations (*Kay et al., 2013b*) and quantified the performance of a flat-response model that predicts the same response level for each data point.

## Template model
### Basic model description
The *Template model* specifies the stimulus properties that drive bottom-up responses in VTC. The model accepts as input a grayscale image and produces as output the predicted response in VWFA and FFA during the fixation task. In brief, the model processes the image using a set of V1-like Gabor filters and then computes a normalized dot product between filter outputs and a category template. The category template can be viewed as capturing the prototypical image statistics of a word (VWFA) or face (FFA). The Template model makes no claim as to how the brain might develop category templates; they might be genetically hard-wired (*Kanwisher, 2010*) or arise from experience with the environment (*Gauthier et al., 1999*). The central claim is that the bottom-up information computed by VTC is, at least to a first approximation, the output of a template operation applied to the stimulus.

The Template model is related to our previously developed Second-order contrast (SOC) model (*Kay et al., 2013c*). Similar to the SOC model, the Template model has a cascade architecture involving two stages of filtering, rectification, and normalization. The first stage of the Template model is taken directly from the SOC model, and the properties of this stage (for example, filter design) were not tweaked to fit the data. The main difference between the Template and SOC models is that the Template model incorporates a specific second-stage filter (the template), whereas the SOC model uses a variance-like operation in the second stage that captures generic sensitivity to second-order contrast. Whether the Template model captures certain response properties, such as invariance to font in VWFA (*Dehaene and Cohen, 2011*) or coarse luminance-contrast selectivity in FFA (*Ohayon et al., 2012*), is an empirical question that can only be resolved through quantitative evaluation on experimental data. For example, our measurements indicate that VWFA responds strongly to polygons (*Figure 1d*); the Template model already accounts for this effect (*Figure 2b*).

### Model details
The first stage of the Template model involves computing a V1-like representation of the image. The image is first resized to 250 pixels × 250 pixels, and luminance values are mapped to the range [−0.5,0.5], which has the effect of mapping the gray background to 0. The model then calculates V1 energy in the same way as the SOC model (*Kay et al., 2013c*). Specifically, the image is projected onto a set of isotropic Gabor filters occurring at eight orientations, two quadrature phases, and a range of positions (63 *x*-positions × 63 *y*-positions). Filters are constructed at a single scale with a peak spatial frequency tuning of 4 cycles per degree (see *Figure 2d*) and a spatial frequency bandwidth of 1 octave (full-width at half-maximum of the amplitude spectrum). Filters are scaled such that filter responses to full-contrast optimal sinusoidal gratings are equal to one. Outputs of quadrature-phase filters are squared, summed, and square-rooted, analogous to the complex-cell energy model (*Adelson and Bergen, 1985*).

After computing V1 energy, the model applies divisive normalization (*Heeger, 1992*), again analogous to the SOC model. The output of each filter is divided by the average output across filter orientations at the same position:

$$ncc_{pos,or} = \frac{(cc_{pos,or})^r}{s^r + \left(\frac{\sum\limits_{or} cc_{pos,or}}{numor}\right)^r} \tag{1}$$

where $ncc_{pos,or}$ is the normalized filter output at a given position and orientation, $cc_{pos,or}$ is the filter output at a given position and orientation, $numor$ is the total number of orientations, and $r$ and $s$ are parameters that control the strength of the normalization. For simplicity and to reduce the potential for overfitting, we do not fit $r$ and $s$ but simply use $r = 1$ and $s = 0.5$, values determined from our previous study (*Kay et al., 2013c*).

At this point in the model, the representation of the image is a 3D matrix of dimensions 63 $x$-positions $\times$ 63 $y$-positions $\times$ 8 orientations. To visualize this representation, a hue-saturation-value image is used (see *Figure 2a*). For each position, a set of 8 vectors is constructed with vector angles corresponding to the filter orientation and vector lengths corresponding to normalized filter output. These vectors are averaged and an image pixel is used to summarize the result. Specifically, the hue of a pixel indicates the angle of the vector average and the value of the pixel indicates the length of the vector average.

The second stage of the Template model involves taking the V1-like representation of the image and comparing it to a category template to generate the predicted response. Specifically, the response is computed as

$$RESP = a \times \frac{|S \cdot T - b\bar{S}|^+}{c + \bar{S}} \tag{2}$$

where $RESP$ is the predicted response, $S$ is the 3D matrix with the V1-like representation of the image, $T$ is the category template, $\bar{S}$ is the average of the elements in $S$, $||^+$ indicates positive half-wave rectification, and $a$, $b$, and $c$ are free parameters (three free parameters).

There are three basic steps in *Equation 2*. The first step is a filtering operation, accomplished by computing the dot product between the stimulus and the template ($S \cdot T$). Intuitively, this operation quantifies the similarity between the stimulus and the template. The second step is the subtraction of average stimulus energy ($-b\bar{S}$) with a free parameter controlling the strength of the subtractive normalization. This subtraction can be interpreted as penalizing non-specific energy in the stimulus, thereby inducing preference for stimulus energy that conforms to the category template. (An alternative interpretation is that the subtraction provides flexibility with respect to the overall mean of the template.) To ease interpretation and ensure that negative responses are not obtained, the result of the subtraction is positively rectified ($||^+$). The third step is division by average stimulus energy ($/ (c + \bar{S})$) with a free parameter controlling the strength of the divisive normalization. This division penalizes non-specific energy in the stimulus, similar to subtractive normalization, but induces a different response geometry (*Zetzsche et al., 1999*). In summary, *Equation 2* computes a dot product between the stimulus and the template that is normalized subtractively and divisively by the average stimulus energy.

Where does the category template in *Equation 2* come from? Given that we do not have sufficient sampling of stimuli to directly estimate templates from the data (but see *Figure 2—figure supplement 1*), we adopted the simple strategy of constructing templates from our stimulus set. Specifically, we took the WORD and FACE stimuli at 100% contrast and used the first stage of the Template model to compute a V1-like representation of these stimuli. This produced for each category, ten points in a 63 $\times$ 63 $\times$ 8 = 31,752 dimensional space. We then computed the centroid of the ten points, producing a category template (example shown in *Figure 2a*). Because the category template is constructed from the same stimuli used in our experiment, it is guaranteed that the Template model predicts large responses to the preferred category (for example, using a category template constructed from the face stimuli guarantees that the face stimuli produce large responses from the model). However, there is no guarantee that the model will accurately account for responses to the other stimuli used in our experiment.

## Model fitting

The Template model was fit to the fixation responses of VWFA and FFA. Model outputs were calculated for all ten images associated with a given stimulus type and then averaged to obtain the final model output for that stimulus type. To aid model fitting, the $S \cdot T$ and $\bar{S}$ quantities were pre-computed and pre-conditioned by dividing each quantity by the mean of that quantity across stimuli. After pre-conditioning, a variety of initial seeds for $b$ and $c$ were evaluated in order to avoid local minima. Specifically, we performed optimization starting from initial seeds corresponding to every combination of $b$ and $c$, where $b$ is chosen from {0 .5 1 1.5 2 3 5} and $c$ is chosen from {.01 .05 .1 .5 1 5 10}.

## Alternative models

(1) The *Category model* predicts a fixed response level for stimuli from the preferred stimulus category (word for VWFA, face for FFA) and a different response level for all other stimuli (two free parameters, one for each response level). Category judgments provided by the subjects were used to determine category membership; for example, words and faces at 0% and 25% phase coherence were reported by subjects as 'other', and are hence not considered to be words and faces by the Category model. (2–3) We evaluated simplified versions of the second-stage normalization used in the Template model. One version, *Template model (only subtractive normalization)*, omits the divisive normalization and thus characterizes responses as a simple linear function of V1-like normalized filter outputs (two free parameters, $a$ and $b$), whereas the other version, *Template model (only divisive normalization)*, omits the subtractive normalization (two free parameters, $a$ and $c$). (4) In *Template model (omit first stage)*, the first stage of the model is omitted and the template operation is performed on a pixel representation of the image, that is, $S$ refers to the original image instead of the V1-like representation of the image (three free parameters, $a$, $b$, and $c$). (5–7) We evaluated the effect of using different templates in the Template model (each model has three free parameters, $a$, $b$, and $c$). *Template model (non-selective template)* uses a template consisting of all ones. *Template model (mixed template)* uses a template generated by unit-length normalizing both the word and face templates and then averaging the templates together. *Template model (random template)* uses a template generated by drawing uniform random values from the range [0,1].

## IPS-scaling model

### Basic model description

The *IPS-scaling model* predicts top-down modulation of VTC by taking into account measurements of IPS activity. The model accepts as input the response in VTC (either VWFA or FFA) during the fixation task and the response in IPS during the stimulus-directed tasks (categorization, one-back), and produces as output the predicted response in VTC during the stimulus-directed tasks. Intuitively, the model answers the question: how much is the bottom-up response in VTC enhanced by the IPS when the subject performs a task on the stimulus? The model can be viewed as a formal implementation of the concept of stimulus-specific scaling (schematized in *Figure 3b*, lower right). Similar ideas regarding top-down scaling induced by the IPS can be found in previous work (*Kayser et al., 2010b*).

### Model details

The IPS-scaling model multiplies the bottom-up response in VTC measured during the fixation task by a scaled version of the IPS response observed during a stimulus-directed task:

$$VTC_{task} = VTC_{bottom} \times (a \cdot IPS_{task} + b) \tag{3}$$

where $VTC_{task}$ is the predicted response in $VTC$ during the stimulus-directed task, $VTC_{bottom}$ is the bottom-up response in $VTC$, $IPS_{task}$ is the response in $IPS$ during the stimulus-directed task, and $a$ and $b$ are parameters that allow a scale and offset to be applied to the IPS response (two free parameters). The final scaling factor that is applied to $VTC_{bottom}$ is shown in *Figure 5b*. The measurements of $IPS$ activity used in the model are extracted using a broad anatomical mask of the $IPS$ (see *Region-of-interest (ROI) definition*).

## Model fitting

The IPS-scaling model was fit to the fixation, categorization, and one-back responses observed in VWFA and FFA. Leave-one-out cross-validation was performed by systematically leaving out each of the categorization and one-back responses. Since measurement noise is present in the fixation responses, treating the fixation responses as exact estimates of bottom-up responses would result in suboptimal model performance (especially in the case of bottom-up responses that are near zero). We therefore devised a procedure that allows flexibility in estimating bottom-up responses (see light lines in *Figure 5b*). In the procedure, a separate parameter is used to model the bottom-up response associated with each stimulus. During model fitting, these bottom-up parameters are initially set to be equal to the measured fixation responses, parameters of the model excluding the bottom-up parameters are optimized, and then all parameters are optimized simultaneously. This procedure was also used for the alternative models described below. Note that the IPS-scaling model uses flexible parameters to accommodate bottom-up stimulus selectivity and does not attempt to characterize the image-processing computations that underlie bottom-up responses (such computations are in the purview of the Template model).

## Alternative models

(1) The *Task-invariant model* posits that top-down modulation does not occur and that a fixed set of responses can characterize all three tasks (zero free parameters). (2–5) We evaluated several phenomenological models for purposes of comparison. The *Additive model* (schematized in *Figure 3b*, upper left) predicts responses during stimulus-directed tasks by adding a constant to bottom-up responses (one free parameter). The *Scaling model* (schematized in *Figure 3b*, lower left) predicts responses during stimulus-directed tasks by multiplying bottom-up responses by a constant (one free parameter). The *Additive model (task-specific)* and *Scaling model (task-specific)* are identical to the previous two models, except that separate constants are used for the categorization and one-back tasks (two free parameters). (6) The *Area-specific enhancement model* (schematized in *Figure 3b*, upper right) is identical to the Scaling model (task-specific) except that scaling is applied only to the stimuli preferred by a given area, that is, words in VWFA and faces in FFA (two free parameters). (7) The *IPS-additive model* predicts responses during stimulus-directed tasks by adding a scaled version of the IPS response to bottom-up responses in VTC (two free parameters). (8–9) To assess the specificity of the IPS enhancement, we evaluated variants of the IPS-scaling model. In the *IPS-scaling (shuffle)* model, IPS responses are shuffled across stimuli and tasks (restricted to the stimulus-directed tasks) before being used in the model (two free parameters). In the *IPS-scaling (shuffle within task) model*, IPS responses are shuffled across stimuli but not across tasks before being used in the model (two free parameters).

## Drift diffusion model

### Basic model description

The *Drift diffusion model* specifies the decision-making operations that underlie performance of the categorization task, and is based upon past research on perceptual decision-making (*Shadlen and Newsome, 2001*; *Heekeren et al., 2004*; *Gold and Shadlen, 2007*; *Kayser et al., 2010a*). The model accepts as input fixation responses in VTC and produces as output predicted reaction times and IPS responses for the categorization task. The basic idea is that VTC responses provide evidence regarding which stimulus category has been presented to the subject, and this evidence is accumulated over time by the IPS in order to make a final decision regarding stimulus category.

### Model details

First, we collect fixation responses in hV4, VWFA, and FFA and divide each set of responses by their mean. This normalization ensures that different ROIs have similar units. Then, for each stimulus category (word, face, other), we compute the centroid of the fixation responses associated with that category, interpret this centroid as a vector, and normalize the vector to unit length. This procedure generates category vectors, defined in a three-dimensional neural space, that point in the directions of the 'arms' of the manifold of the VTC representation (see *Figure 3e*).

Next, we take the VTC fixation response for a given stimulus and project this response onto the category vector associated with that stimulus. The working hypothesis is that this operation is

performed by neurons in IPS and that the magnitude of the projection indicates the strength of evidence for that specific category. For example, there might be an IPS neuron that receives information from VTC and responds strongly when the VTC response is consistent with the category vector corresponding to a word.

In accordance with drift diffusion models, we posit that evidence is accumulated until a threshold is reached, at which point the decision is made. This generates a prediction of the reaction time required to perform the categorization task on the stimulus:

$$RT = c + \frac{b}{R \cdot V} \qquad (4)$$

where $RT$ is the predicted reaction time, $R$ is the VTC fixation response, $V$ is the category vector associated with the stimulus, $b$ is a parameter that controls the threshold, and $c$ is a parameter that compensates for non-decision time (for example, motor response) (two free parameters). $R \cdot V$ is interpreted as a drift rate, and $b/(R \cdot V)$ is the time required to reach the threshold (see *Figure 6a*). Note that our instantiation of the drift diffusion model is relatively simple, as it is non-stochastic and does not characterize trial-to-trial variability. Thus, it can be viewed as a simplified version of the classic drift diffusion model (*Ratcliff, 1978*). Also, being non-stochastic, our model bears similarity to the linear ballistic accumulator model (*Brown and Heathcote, 2008*), a model that also uses the idea of evidence accumulation (see *Donkin et al., 2011* for discussion of these different models).

Given that neuronal responses in parietal cortex reflect the duration of the decision-making process (*Shadlen and Newsome, 2001*), we can use RT to predict IPS activity. A detailed model relating RT to BOLD measurements of IPS activity requires precise characterization of neural dynamics during decision-making and IPS subdivisions that might represent evidence accumulation for different stimulus categories. For the purposes of this study, we use a simple model that posits a monotonically increasing relationship between RT and the IPS response:

$$IPS = a \times \tanh(b \cdot RT + c) + d \qquad (5)$$

where $IPS$ is the predicted $IPS$ response, $RT$ is the observed reaction time for a given stimulus, $\tanh$ is the hyperbolic tangent function intended as a generic sigmoidal nonlinearity, and $a$, $b$, $c$, and $d$ are free parameters (four free parameters).

## Alternative models

(1) The *Drift diffusion model (separate thresholds)* uses a separate threshold parameter for each stimulus category (four free parameters, one for non-decision time and three for thresholds). This allows us to assess the validity of having a single threshold parameter in the model. (2) The *Drift diffusion model (axis-aligned category vectors)* uses category vectors that are aligned with the axes of the multi-dimensional neural space (four free parameters, similar to the previous model). For example, in this model, the word category vector is a vector that is one along the VWFA axis and zero along the hV4 and FFA axes. This model tests the idea that evidence for words and faces is contributed only by the VTC regions selective for those categories.

## Additional wide-range-of-stimuli dataset

### Experimental design

To assess the generalization performance of the Template model, we collected an additional dataset involving a wider range of stimuli. This dataset was collected from one subject (an author; male; age 34). Informed written consent was obtained, and the protocol was approved by the University of Minnesota Institutional Review Board. The experiment was similar in design to the main experiment. Stimuli included 22 images from the main experiment (one image from each of the 22 stimulus types), 92 images from a previous study investigating object representation in ventral temporal cortex (*Kriegeskorte et al., 2008*), and 19 other images not used in this paper. As in the main experiment, images were approximately 2° × 2° in size. In each 4 s trial, a single image was flashed using an 800 ms ON, 200 ms OFF duty cycle. During a run, each image was presented in one trial, and 11 blank trials were also included. Each run lasted 608 s (10.1 min), and a total of 10 runs were collected. During stimulus presentation, the subject performed a variant of the fixation task. A small dot (0.1°×0.1°) was present at the center of the display and switched to one of five shades of red

(ranging from (40,0,0) to (255,0,0) in five equally spaced increments) every 1200 ms (repetitions allowed). The subject was instructed to press a button whenever the luminance of the central dot increased and a different button whenever the luminance decreased.

## MRI data acquisition

MRI data were collected at the Center for Magnetic Resonance Research at the University of Minnesota using a 7T Siemens Magnetom scanner and a custom 4-channel-transmit, 32-channel-receive RF head coil. Stimuli were presented using a Cambridge Research Systems BOLDscreen 32 LCD monitor (resolution 1920 × 1080 at 120 Hz; viewing distance 189.5 cm). Functional data were acquired using 84 oblique slices covering occipitotemporal cortex: slice thickness 0.8 mm, slice gap 0 mm, field-of-view 160 mm (FE) × 129.6 mm (PE), phase-encode direction inferior-superior. A T2*-weighted, single-shot, gradient-echo EPI sequence was used: matrix size 200 × 162, TR 2.2 s, TE 22.4 ms, flip angle 80°, phase partial Fourier 6/8, in-plane acceleration factor (iPAT) 3, slice acceleration factor (multiband (*Moeller et al., 2010*)) 2, nominal spatial resolution 0.8 mm × 0.8 mm × 0.8 mm.

## Data analysis

After pre-processing, the functional data were averaged across the thickness of gray matter and then analyzed using GLMdenoise (as in the main experiment). Beta weights extracted from FFA and VWFA were then modeled using several variants of the Template model. Model accuracy was quantified using 20-fold cross-validation (random subsets of the stimuli). (1) The *Template model (original)* is the same model used in the main experiment (three free parameters). Importantly, the category template in the model is fixed and not adjusted to the new dataset. (2) The *Template model (half-max average)* is a simple extension of the Template model in which the category template is estimated as follows: on each cross-validation iteration, using only the training set, identify responses that are at least half of the maximum response and then compute the centroid (in the V1-like representation) of the stimuli corresponding to these responses (three free parameters plus nonparametric fitting of the template). Note that this procedure is cross-validated in the sense that the category template is fit only to the training set; whether the estimated category template generalizes to novel stimuli is an empirical question that is assessed through cross-validation. (3) The *Template model (half-max cluster)* further extends the model to accommodate multiple category templates. The logic is that just as V1 models use filters at multiple orientations and spatial scales to characterize the overall V1 response, we might conceptualize FFA and VWFA as containing multiple templates tuned to different types of stimuli (for example, different face viewpoints, different fonts). First, we identify responses that are at least half of the maximum response from the training set. We then perform $k$-means clustering (in the V1-like representation) on the stimuli corresponding to these responses. We use a cosine metric to quantify distance, and we select the solution that minimizes cluster assignment error across 100 random initializations of centroid positions. The obtained centroids are unit-length normalized and then used as category templates in the Template model. For simplicity and to avoid overfitting, we compute the predicted response as a simple sum across the independent responses of different category templates and use the same parameter values ($a$, $b$, $c$ in *Equation 2*) for different templates. We systematically vary the number of clusters from 1 through 8, and select the number that maximizes cross-validation performance (four free parameters—three for $a$, $b$, and $c$, one for the number of clusters—plus nonparametric fitting of templates).

## Code availability

Software code implementing the model proposed in this paper is available at http://cvnlab.net/vtcipsmodel/.

## Acknowledgements

We thank K Grill-Spector for providing the face and house stimuli used in the main experiment, R Kiani and N Kriegeskorte for providing the object stimuli used in the retinotopic mapping experiment, A Vu and E Yacoub for collecting pilot data, C Gratton, M Harms, and L Ramsey for scanning assistance, and K Weiner for assistance with ROI definition. We also thank P Elder, C Gratton, S

Petersen, A Rokem, A Vogel, and J Winawer for helpful discussions. This work was supported by the McDonnell Center for Systems Neuroscience and Arts and Sciences at Washington University (KNK) and NSF Grant BCS-1551330 (JDY). Computations were performed using the facilities of the Washington University Center for High Performance Computing, which were partially provided through grant NCRR 1S10RR022984-01A1.

## Additional information

### Funding

| Funder | Grant reference number | Author |
|---|---|---|
| McDonnell Center for Systems Neuroscience | | Kendrick N Kay |
| Washington University in St. Louis | | Kendrick N Kay |
| National Science Foundation | BCS-1551330 | Jason D Yeatman |

The funders had no role in study design, data collection and interpretation, or the decision to submit the work for publication.

### Author contributions

KNK, Wrote the paper, Designed the experiments, Conducted the experiments, Analyzed the functional and behavioral data; JDY, Wrote the paper, Designed the experiments, Analyzed the diffusion data

### Author ORCIDs

Kendrick N Kay, http://orcid.org/0000-0001-6604-9155
Jason D Yeatman, http://orcid.org/0000-0002-2686-1293

### Ethics

Human subjects: Informed written consent was obtained from all subjects, and the experimental protocol was approved by the Washington University in St. Louis Institutional Review Board and the University of Minnesota Institutional Review Board.

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
