## [Decision Letter]

Thank you for submitting your article "Bottom-up and top-down computations in word- and face-selective cortex" for consideration by *eLife*. Your article has been reviewed by three peer reviewers, and the evaluation has been overseen by a Reviewing Editor and David Van Essen as the Senior Editor. The following individuals involved in review of your submission have agreed to reveal their identity: Floris de Lange (Reviewer #1); Geoffrey K. Aguirre (Reviewer #2).

The reviewers have discussed the reviews with one another and the Reviewing Editor has drafted this decision to help you prepare a revised submission.

Summary:

This study used fMRI and computational modeling to investigate bottom-up and top-down contributions to activation patterns in the ventral temporal cortex (VTC) while human subjects viewed different stimuli (face or text of different contrasts and phase coherences) under different task conditions (fixation, one-back, and categorization). They show that responses in human FFA and VWFA are strongly influenced by both stimulus properties and task demands. They also identify a region of the IPS as a source of attentional modulation of the VTC. Finally, they propose a template-matching model that can reasonably account for the observed bottom-up and top-down activity fluctuations in VTC. Given that analyses of these types of experimental data have typically been framed more qualitatively, the reviewers agree that a quantitative and rigorous modeling approach is certainly a welcome contribution. The model also is noted as having the virtue of simplicity.

Nonetheless, the reviewers were also in agreement that there are a number of major issues that need to be addressed, described below.

Essential revisions:

1) Scope and readability: This paper contains an ambitious number of ideas. A challenge for the reader is that the model details are the primary product of the work, yet most of these details are relegated to Methods or Supplementary Materials. Here are some suggestions:

a) Consider moving Figure 2 to the supplement. This is a nice result, but it is not needed for the subsequent modeling work.

b) Re-arrange the results to present: 1) "VTC responses depend on both stimulus properties and cognitive task", 2) "Model of bottom-up computations in VTC". Observe that this model accounts for the fixation data but not the task data, 3) "IPS is the source of top-down modulation to VTC", 4) "Model of top-down computations in VTC", 5) "Model of perceptual decision-making in IPS".

c) Provide some detail in the main text regarding each model. In particular, the primary features of the model that impacts inference should be described. For example, the fact that the category template model is generated from the stimuli themselves is quite relevant.

d) Move the DTI results to the supplement or remove. It is certainly worthwhile noting that the VTC and IPS are connected by a white matter pathway. However, Jason's prior work demonstrates the existence of this fiber tract, and the current paper simply re-demonstrates the pathway. In the absence of some (e.g.) individual difference measurement or other use of the data to support inferences beyond the mere existence of the pathway, this result is not particularly additive.

2) Connection to literature of top-down effects on contrast-sensitivity. The paper ignores a large body of work that has also tried to formalize top-down affects such as attention on contrast-sensitivity in early visual cortex of humans and monkeys. This literature has described contrast-gain, response-gain, and additive-offsets of contrast sensitivity with top-down modulation due to attention. Some connection should be made between these results and this vast literature.

Moreover, how can the results be reconciled with findings from other groups that areas such as EBA and FFA also have category-selective responses in congenitally blind people? Does it pose a problem for their framework?

3) Model analysis and comparison.

a) It would be useful to more completely explain the model comparisons, and the work that the various parameter of each model perform in fitting the data. For example, the category template model provides good fits to the fixation-response data across contrast and phase coherence. How does these fits depend on the Gabor filter bank, the category template, and/or the parameters available in the normalization step? Moreover, the authors conclude that a scaling mechanism is in place for amplification of VTC responses. The alternative models that they draw in Figure 2 seem to make qualitatively quite similar predictions. How do they arrive at the conclusion of which mechanism is most supported by the data? Formal model comparison would be useful. Finally, why don't the top-down and drift-diffusion models make use of the category template model that is fit to the fixation data? This seems like a missed opportunity. Is this because the model fits are poor or a technical limitation?

b) How general is the model for categorical representation in FFA and VWFA? The model transforms images into weightings of a V1-like representation (gabor wavelet pyramid) and then projects these onto a template for faces and text. This amounts to little more than putting a linear classifier on a V1 like representation, which would seem much too simplistic to account for known properties of invariance in high-order visual cortex to things like view angle, lighting, rotation, scale, etc. Moreover, how well can the model predict responses to non-template stimuli? Because the templates are built as the centroid of responses of the actual stimuli used, it is not surprising that it can explain the face and word responses. As the authors point out, the more important tests are for stimuli outside of those that the model is built on. In Figure 4 the phase coherence manipulation (as well as noise and some alternative stimuli) are relevant. However, the phase coherence manipulation in the data of Figure 4 look more like a step function then what the model predicts, suggesting poor fits. Moreover, the supplementary data show that the only way to save the template model for a more general set of stimuli is to basically build the templates based on what stimuli evoked half-maximal response, which is like tweaking the templates to match the responses. While the legend mentions train and test sets, which would suggest cross-validation that might alleviate some of these concerns, the details are omitted. If train and test are different sets of responses to the same presentations of images, then the cross-validation would say more about the consistency in response then whether the model generalizes to new stimuli.

c) It seems a bit unfair to compare the parameterized category template model with a uniform category model. The category model has no contrast-sensitivity and thus gives a flat response across contrast in Figure 4. How much difference would there be between models, if you just allowed the category model to be contrast-sensitive? Likewise, the category model does seem to step for phase coherence, but no description was given for this; is this because at the lower phase coherences the stimulus category cannot be discerned? Also, how is it that some of the category model fits show some decrease in response as a function of contrast for the preferred category? In general, it would be useful to move some of the supplementary material that describes the performance of other models (including for the RT data) to the main text, since the model comparisons are of central importance to this study.

4) IPS fit circularity. Despite the assurances in Methods, the top-down IPS model faces a bit of a circularity problem. The IPS region was found using a search process that sought signals that could account for task effects. It is therefore not surprising to learn (Figure 4, bottom) that these signals well fit the data. Indeed, the r=0.96 fit is a bit worrying in this regard and suggests the possibility of over-fitting. Here are some options to rectify: a) Explicitly describe the fitting as a demonstration exercise, not to be taken as independent evidence of the role of the IPS; b) Obtain data from a new set of subjects, using the IPS region identified in the initial set of subjects to replicate the result.

5) Consistency of model framing of parietal cortex. The parietal cortex is first modeled as the source of an attention signal that modulates VTC, and then also as an accumulator of evidence from VTC. Is this plausible, and can the fMRI data support this conclusion? For example, given high task difficulty, does strong activation of IPS imply strong neural activation to provide attentional control or weak but temporally extended neural activation to accumulate weak evidence?

6) Parietal cortex correlation analysis. The suggested effect of attention is posited to be a gain effect – yet when looking for the source of this gain signal, the fixation condition is subtracted off. If it really is a gain, subtraction would be expected to leave some stimulus-driven effect in the other conditions, and correlations to parietal cortex could be due in part to this residual stimulus representation.

7) Correlation is not causation. A correlation with parietal cortex and VTC modulation is reported and interpreted as a causative signal from IPS areas. But the analysis is a correlation; it does not prove which way causation goes.

[Editors' note: further revisions were requested prior to acceptance, as described below.]

Thank you for resubmitting your work entitled "Bottom-up and top-down computations in word- and face-selective cortex" for further consideration at *eLife*. Your revised article has been favorably evaluated by David Van Essen (Senior editor) and a Reviewing editor.

The manuscript has been improved but there are some remaining issues that need to be addressed before acceptance, as outlined below:

1) The new discussion that makes connections to existing literature on top-down effects on contrast sensitivity is a welcome addition (Discussion section paragraph four). It would likely benefit readers to ascribe specific references to the various models discussed.

2) The cross-validation model comparisons could also be described better. For example, Figure 5 shows 10 models + 2 benchmarks in the fig, but only 7 are described (briefly) in the text. It would also be useful to describe the number of free parameters in each model. How do goodness of fits compare when taking into account the different degrees of freedom?

3) Figure 5: please indicate meaning of colors, arrows, etc. in the legend, not just the associated text.

4) DDM (Figure 6): It would be helpful to readers to clarify the terminology related to the decision model. A DDM has a single decision variable that has Brownian-like dynamics; here the model appears to be closer to a race between linear ballistic accumulators (three in the case of the 3AFC categorization task). It also would be useful to discuss these results in the context of a host of findings that have largely highlighted how difficult it is to use slow BOLD signals to make inferences about accumulator-like activity occurring over relatively short timescales, as in this study (see a nice discussion of these issues in Krueger et al., 2017, "Evidence accumulation detected in BOLD signal using slow perceptual decision making," J Neurosci Methods).

---

## [Author Response]

*Essential revisions:*

*1) Scope and readability: This paper contains an ambitious number of ideas. A challenge for the reader is that the model details are the primary product of the work, yet most of these details are relegated to Methods or Supplementary Materials. Here are some suggestions:*

*a) Consider moving Figure 2 to the supplement. This is a nice result, but it is not needed for the subsequent modeling work.*

*b) Re-arrange the results to present: 1) "VTC responses depend on both stimulus properties and cognitive task", 2) "Model of bottom-up computations in VTC". Observe that this model accounts for the fixation data but not the task data, 3) "IPS is the source of top-down modulation to VTC", 4) "Model of top-down computations in VTC", 5) "Model of perceptual decision-making in IPS".*

*c) Provide some detail in the main text regarding each model. In particular, the primary features of the model that impacts inference should be described. For example, the fact that the category template model is generated from the stimuli themselves is quite relevant.*

*d) Move the DTI results to the supplement or remove. It is certainly worthwhile noting that the VTC and IPS are connected by a white matter pathway. However, Jason's prior work demonstrates the existence of this fiber tract, and the current paper simply re-demonstrates the pathway. In the absence of some (e.g.) individual difference measurement or other use of the data to support inferences beyond the mere existence of the pathway, this result is not particularly additive.*

We appreciate the suggestions, and we are aware that our work is technically dense. We wholeheartedly agree that the model details are critical; it is always challenging to balance details against brevity in this type of work.

We think Figure 2 is quite important—it is a nice way of understanding the structure of the data, and is independent of the modeling effort. The revised introductory paragraph of the corresponding section ‘Top-down modulation acts as a stimulus-specific scaling’ makes clear that analysis sets the stage for the subsequent modeling effort.

Also, we think the DTI results are an important and necessary precursor for building a model that posits a direct relationship between signals measured in two spatially separated brain regions (VTC and IPS). Previous work did not examine the relationship between VOF endpoints and functionally defined FFA ROIs in individual subjects. We believe that establishing the spatial proximity of the FFA and VWFA to VOF fibers that connect to the functionally defined IPS hotspot demonstrates that the model in the next section is anatomically plausible.

We re-arrange the results, use better subheadings, and expand introductory paragraphs, as per suggestion (b). We add detail regarding the various models in the main text, as per suggestion (c). We also merge the model comparison results into the main text, which further fleshes out the modeling work. These changes are located throughout the body of the main text as well as in Figure 2, Figure 5 and Figure 6.

*2) Connection to literature of top-down effects on contrast-sensitivity. The paper ignores a large body of work that has also tried to formalize top-down affects such as attention on contrast-sensitivity in early visual cortex of humans and monkeys. This literature has described contrast-gain, response-gain, and additive-offsets of contrast sensitivity with top-down modulation due to attention. Some connection should be made between these results and this vast literature.*

We agree; we had a section discussing this literature in an early version of the paper, but that section was removed for length reasons.

We discuss the literature on top-down effects on contrast-sensitivity (Discussion section paragraph four).

*Moreover, how can the results be reconciled with findings from other groups that areas such as EBA and FFA also have category-selective responses in congenitally blind people? Does it pose a problem for their framework?*

It is certainly possible that there exist a variety of types of top-down signals, and we are open to the possibility that our current model is incomplete. For example, we might suspect that the VWFA can receive top-down signals from language areas but that these signals do not play a large role in the three tasks sampled in our experiment. With further experimental measurements, our framework could be expanded to accommodate additional types of top- down signals. This line of thinking is discussed in the paper (subsection “Model of top-down computations in VTC”).

*3) Model analysis and comparison.*

*a) It would be useful to more completely explain the model comparisons, and the work that the various parameter of each model perform in fitting the data. For example, the category template model provides good fits to the fixation-response data across contrast and phase coherence. How does these fits depend on the Gabor filter bank, the category template, and/or the parameters available in the normalization step?*

This is a good question. The Gabor filter bank was taken identically from the previous computational model that we developed (SOC model). Thus, there was no tweaking of the filter bank to fit the data, thereby avoiding overfitting. Exploring the dependence of the model on the filter bank will require additional measurements and will be the focus of future work. As for the category template and normalization step, these were indeed investigated, but results were presented in the supplement.

We merge the model comparisons (previously in the supplement) into the main text. This shows how the category template and the normalization step affect model performance (subsection “Model of bottom-up computations in VTC”; Figure 2), and shows a brief investigation of spatial frequency properties (Figure 2). Also, we revise the text to explicitly mention that the filter bank is taken directly from the previous SOC model (subsection “Template model”).

*Moreover, the authors conclude that a scaling mechanism is in place for amplification of VTC responses. The alternative models that they draw in Figure 2 seem to make qualitatively quite similar predictions. How do they arrive at the conclusion of which mechanism is most supported by the data? Formal model comparison would be useful.*

Formal model comparison is performed; we apologize that this was not clear in the original version of the paper.

We implement the ‘Area-specific enhancement’ model (formal implementation of this model was not present in the original paper but now exists in Figure 5). We also alert the reader that formal evaluation of models is performed (subsection “Top-down modulation acts as a stimulus-specific scaling”).

*Finally, why don't the top-down and drift-diffusion models make use of the category template model that is fit to the fixation data? This seems like a missed opportunity. Is this because the model fits are poor or a technical limitation?*

We appreciate the reviewer’s attention to detail. It is not a technical limitation. Rather, given that the model fits are good but not perfect, we decided to isolate the performance evaluation of each component of the model (Template, IPS-scaling, Drift-diffusion). This approach allows us to cleanly assess the quality of each component. For example, if the Template model had been used in the implementation of the IPS-scaling model, then it would not be clear whether shortcomings in cross-validation performance of the IPS-scaling model were due to inaccuracies in the Template model or due to actual inaccuracies in the modeling of top-down modulation.

We revise the manuscript to address and discuss this issue (subsection “Computational modelling”).

*b) How general is the model for categorical representation in FFA and VWFA? The model transforms images into weightings of a V1-like representation (gabor wavelet pyramid) and then projects these onto a template for faces and text. This amounts to little more than putting a linear classifier on a V1 like representation, which would seem much too simplistic to account for known properties of invariance in high-order visual cortex to things like view angle, lighting, rotation, scale, etc.*

It may very well be the case that high-level visual cortex has complex processing dedicated to achieving invariances, and it may seem that certain models are too simplistic to implement such behaviors. But our view is that the precise numbers matter and that there is no choice but to collect appropriate experimental data and perform quantitative modeling work to see to what extent a particular model can account for the data. As a case in point, in previous work we explored the issue of position and size invariance and found that, perhaps surprisingly, a relatively simple computational mechanism can account for experimentally measured effects (Kay et al., 2013).

We believe that the field needs increased effort towards developing general, predictive, and interpretable models that are directly validated against empirical data. Even though our current model does not perfectly account for all possible experimental manipulations, we believe there is value in starting somewhere and defining a framework that can be refined in future studies. A logical extension of the present work is to extend the model to address spatial invariance/tolerance, given our previous work on this topic (Kay et al., Current Biology, 2015). Investigating and validating an extended model will require a larger and more extensive stimulus set.

*Moreover, how well can the model predict responses to non-template stimuli? Because the templates are built as the centroid of responses of the actual stimuli used, it is not surprising that it can explain the face and word responses. As the authors point out, the more important tests are for stimuli outside of those that the model is built on.*

We agree with the sentiment here: testing generalization is important. For the main set of data, we use cross-validation across stimuli, which provides some assessment of generalization power. We find that the model predicts the response to non-preferred stimuli, meaning that the relative response amplitude in VWFA and FFA depends on the similarity of the visual stimulus to the template. For example, the model predicts the relative VWFA response to polygons, houses, faces, and noise patterns, even though these stimuli were not a part of the word template. The additional larger set of data that we included provides even better assessment of generalization power (for these data, we also used cross-validation across stimuli).

*In Figure 4 the phase coherence manipulation (as well as noise and some alternative stimuli) are relevant. However, the phase coherence manipulation in the data of Figure 4 look more like a step function then what the model predicts, suggesting poor fits.*

Yes, the model does not satisfactorily capture the step-like shape of the phase-coherence responses (as mentioned in the Discussion section paragraph two). This means that there is room for improvement. It is interesting to consider potential paths forward. One route is to posit that there may be some sort of sigmoidal nonlinearity in the second-stage filtering process. A different route is to explore the possibility that during the fixation task, there was some amount of “unavoidable attention to the stimulus”, in the sense that beyond a certain phase-coherence level, the stimulus automatically attracts attention. In this case, the step-like shape might reflect a combination of the true bottom-up response and some accidental top-down modulation.

*Moreover, the supplementary data show that the only way to save the template model for a more general set of stimuli is to basically build the templates based on what stimuli evoked half-maximal response, which is like tweaking the templates to match the responses. While the legend mentions train and test sets, which would suggest cross-validation that might alleviate some of these concerns, the details are omitted. If train and test are different sets of responses to the same presentations of images, then the cross-validation would say more about the consistency in response then whether the model generalizes to new stimuli.*

The reviewer is correct that for the larger set of data in the supplement, we allowed fitting of the templates used in the Template model. The reviewer is also correct that cross-validation is important and that different cross-validation schemes assess different types of generalization. In the present case, the models are cross-validated using train and test sets that reflect different stimuli (as opposed to trials); thus, stimuli in the test set are not involved in template fitting, and modeling procedures genuinely assess how well the models generalize to new stimuli. This information was provided in the Methods section ‘Additional wide-range-of-stimuli dataset’.

To the caption of Figure 2—figure supplement 1, we add some information regarding cross-validation and refer the reader to the appropriate Methods section for details.

*c) It seems a bit unfair to compare the parameterized category template model with a uniform category model. The category model has no contrast-sensitivity and thus gives a flat response across contrast in Figure 4. How much difference would there be between models, if you just allowed the category model to be contrast-sensitive?*

We are not sure why it would be fair to allow contrast sensitivity to the uniform category model. Arguably, the whole point of the category model is that representations might depend solely on the category of a stimulus, not on low-level visual properties, nor the specific exemplar of a category that is presented. It is possible to tack on ad hoc tuning for visual properties to the category model, but we are not sure what insight this would provide.

To be clear, from a conceptual point of view, the Template model is not incompatible with the category model. One can view the Template model as a quantitative elaboration of the category model, showing how category selectivity is generated.

*Likewise, the category model does seem to step for phase coherence, but no description was given for this; is this because at the lower phase coherences the stimulus category cannot be discerned?*

That is correct; the category assignments used in the category model are based on the subjects’ behavioral reports. This is described in the Methods. We acknowledge that the step-like phase coherence response is one aspect of the data that is not accurately predicted by the Template model (see Point 10).

To prevent confusion, we add text specifically addressing the phase coherence cases (subsection “Template model”).

*Also, how is it that some of the category model fits show some decrease in response as a function of contrast for the preferred category?*

What is shown in Figure 2 are cross-validated predictions of the various models using leave-one- stimulus-out cross-validation. Thus, even though the category model is theoretically binary in nature (i.e. response for words vs. response for non-words), because of quantitative model fitting and cross-validation, it is possible for predicted values to be non-binary. This is why the category model exhibits a decrease in predicted response as a function of contrast.

We add some clarification regarding the response decrease (see Figure 2 caption).

*In general, it would be useful to move some of the supplementary material that describes the performance of other models (including for the RT data) to the main text, since the model comparisons are of central importance to this study.*

We agree, and we merge all of the model cross-validation results into the main text (see Figure 2, Figure 5 and Figure 6).

*4) IPS fit circularity. Despite the assurances in Methods, the top-down IPS model faces a bit of a circularity problem. The IPS region was found using a search process that sought signals that could account for task effects. It is therefore not surprising to learn (Figure 4, bottom) that these signals well fit the data. Indeed, the r=0.96 fit is a bit worrying in this regard and suggests the possibility of over-fitting. Here are some options to rectify: a) Explicitly describe the fitting as a demonstration exercise, not to be taken as independent evidence of the role of the IPS; b) Obtain data from a new set of subjects, using the IPS region identified in the initial set of subjects to replicate the result.*

We agree with the reviewer’s concerns (hence the text in the Methods), though we differ in how deleterious the effects are deemed to be. Given that we are not claiming to have strong evidence for precise localization of the top-down region, we prefer the approach of using a liberal anatomical mask over the suggested strategy of functionally identifying the region in one set of subjects and using this region in a separate set of subjects.

We revise the text to explicitly acknowledge that the top-down IPS model should not be taken as independent evidence of the involvement of the IPS (“subsection “Model of top-down computations in VTC”). We also enlarge the maps in Figure 4—figure supplement 1 so that it is easier to examine the localization results.

*5) Consistency of model framing of parietal cortex. The parietal cortex is first modeled as the source of an attention signal that modulates VTC, and then also as an accumulator of evidence from VTC. Is this plausible, and can the fMRI data support this conclusion? For example, given high task difficulty, does strong activation of IPS imply strong neural activation to provide attentional control or weak but temporally extended neural activation to accumulate weak evidence?*

Yes, our model framing implies two views of the parietal cortex. One, parietal cortex is viewed as an accumulator of evidence. Two, it is also viewed as providing top-down modulation of VTC. We see no reason that these two roles cannot co-exist. Of course, this characterization of parietal cortex could be strengthened by additional experimental evidence. In particular, the reviewer alludes to the issue of temporal dynamics, which we agree is very interesting to consider (and is discussed in paragraph five of the Discussion section). We imagine that during high task difficulty, IPS activity is temporally extended to accumulate weak evidence and that this in turns induces temporally extended top-down modulation of VTC.

We revise the caption of Figure 6 in order to clarify the architecture of our model.

*6) Parietal cortex correlation analysis. The suggested effect of attention is posited to be a gain effect – yet when looking for the source of this gain signal, the fixation condition is subtracted off. If it really is a gain, subtraction would be expected to leave some stimulus-driven effect in the other conditions, and correlations to parietal cortex could be due in part to this residual stimulus representation.*

The reviewer is correct that the correlation procedure involves subtraction of fixation responses whereas the top-down effect is more accurately described as a gain-like effect.

The correlation procedure is primarily intended to localize the cortical region involved in the top-down modulation, as opposed to providing a precise quantification of the top-down effect. The advantage of correlation is that it is robust to measurement noise and is computationally efficient. On the other hand, it is technically challenging to produce a robust and efficient method that characterizes multiplicative interactions (such a method would presumably involve some sort of regularization as well as nonlinear optimization for every voxel in the brain).

We point out that quantitative examination of the top-down effect is provided in Figure 5. There, we directly compare multiplicative and additive mechanisms, and it is shown that having the IPS interact multiplicatively (IPS-scaling model) leads to better cross-validated predictions than when it interacts additively (IPS-additive model).

We revise the manuscript to discuss these issues (subsections “Task-based functional connectivity” and “Model of top-down computations in VTC”).

7) Correlation is not causation. A correlation with parietal cortex and VTC modulation is reported and interpreted as a causative signal from IPS areas. But the analysis is a correlation; it does not prove which way causation goes.

We agree with the reviewer.

To improve clarity, we revise the paper to explicitly state that the correlational results do not in themselves prove causation but that we are imposing an interpretation of the results in the context of a computational model (Discussion section).

[Editors' note: further revisions were requested prior to acceptance, as described below.]

*The manuscript has been improved but there are some remaining issues that need to be addressed before acceptance, as outlined below:*

*1) The new discussion that makes connections to existing literature on top-down effects on contrast sensitivity is a welcome addition (Discussion section paragraph four). It would likely benefit readers to ascribe specific references to the various models discussed.*

As suggested, we insert a key reference for each model discussed (Discussion section paragraph four).

*2) The cross-validation model comparisons could also be described better. For example, Figure 5 shows 10 models + 2 benchmarks in the fig, but only 7 are described (briefly) in the text.*

We include additional description of the model comparisons of Figure 5 (subsection “Model of top-down computations in VTC”).

*It would also be useful to describe the number of free parameters in each model. How do goodness of fits compare when taking into account the different degrees of freedom?*

We agree that reporting the numbers of parameters is useful for describing the models, and we clarify this issue in the Methods section.

There are multiple methods for assessing model accuracy. One method, alluded to by the reviewer, is to use simple goodness-of-fit and penalize models based on numbers of free parameters. Another method is to use cross-validation. We have chosen to use cross-validation in our study as it is simple and makes minimal assumptions about the nature of the noise distribution. Since cross-validation already controls for model complexity, the performances of the models in our study can be directly compared.

We include detail on the numbers of parameters for all of the models used in this study. Also, we explicitly discuss the issue of model complexity and cross-validation (subsection “Computational modelling”).

*3) Figure 5: please indicate meaning of colors, arrows, etc. in the legend, not just the associated text.*

We add the requested description to the figure legend.

*4) DDM (Figure 6): It would be helpful to readers to clarify the terminology related to the decision model. A DDM has a single decision variable that has Brownian-like dynamics; here the model appears to be closer to a race between linear ballistic accumulators (three in the case of the 3AFC categorization task).*

We thank the reviewer for this pointer. We appreciate the distinctions now, having reviewed the relevant literature.

We clarify how our model relates to existing models of evidence accumulation (Figure 6 and subsection “Drift diffusion model”).

*It also would be useful to discuss these results in the context of a host of findings that have largely highlighted how difficult it is to use slow BOLD signals to make inferences about accumulator-like activity occurring over relatively short timescales, as in this study (see a nice discussion of these issues in Krueger et al., 2017, "Evidence accumulation detected in BOLD signal using slow perceptual decision making," J Neurosci Methods).*

Given that the BOLD signal can be interpreted as a convolution of a sluggish hemodynamic response function with neural activity dynamics, it can be expected that although relatively small differences in neural activity durations (e.g. ranging from 0–2 s) will not show up very well in the temporal shape of the BOLD response, they will show up in BOLD response amplitudes, as previously noted (Kayser et al., 2010).

We agree that there may be special experimental tricks that can be used to glean information regarding accumulator-like activity in the shape of the BOLD response, as discussed in the paper mentioned by the reviewer.

We explicitly address the issue of neural activity dynamics and its expected impact on the BOLD response (subsection “Model of perceptual decision-making in IPS”). We also insert some additional references to the work of Kayser and colleagues.